# NAS-Bench-Suite-Zero:
# Accelerating Research on Zero Cost Proxies

**Arjun Krishnakumar**[*1]**, Colin White**[*2]**, Arber Zela**[*1]**, Renbo Tu**[*3]**,**
**Mahmoud Safari**[1]**, Frank Hutter**[1,4]

[1]University of Freiburg, [2]Abacus.AI, [3]University of Toronto,
[4]Bosch Center for Artificial Intelligence

## Abstract

Zero-cost proxies (ZC proxies) are a recent architecture performance prediction technique aiming to significantly speed up algorithms for neural architecture search (NAS). Recent work has shown that these techniques show great promise, but certain aspects, such as evaluating and exploiting their complementary strengths, are under-studied. In this work, we create `NAS-Bench-Suite-Zero`: we evaluate 13 ZC proxies across 28 tasks, creating by far the largest dataset (and unified codebase) for ZC proxies, enabling orders-of-magnitude faster experiments on ZC proxies, while avoiding confounding factors stemming from different implementations. To demonstrate the usefulness of `NAS-Bench-Suite-Zero`, we run a large-scale analysis of ZC proxies, including a bias analysis, and the first information-theoretic analysis which concludes that ZC proxies capture substantial complementary information. Motivated by these findings, we present a procedure to improve the performance of ZC proxies by reducing biases such as cell size, and we also show that incorporating all 13 ZC proxies into the surrogate models used by NAS algorithms can improve their predictive performance by up to 42%. Our code and datasets are available at https://github.com/automl/naslib/tree/zerocost.

## 1 Introduction

Algorithms for neural architecture search (NAS) seek to automate the design of high-performing neural architectures for a given dataset. NAS has successfully been used to discover architectures with better accuracy/latency tradeoffs than the best human-designed architectures [5, 9, 28, 38]. Since early NAS algorithms were prohibitively expensive to run [58], a long line of recent work has focused on improving the runtime and efficiency of NAS methods (see [9, 49] for recent surveys).

A recent thread of research within NAS focuses on *zero-cost proxies* (ZC proxies) [1, 23]. These novel techniques aim to give an estimate of the (relative) performance of neural architectures from just a *single minibatch of data*. Often taking just five seconds to run, these techniques are essentially "zero cost" compared to training an architecture or to any other method of predicting the performance of neural architectures [48]. Since the initial ZC proxy was introduced [23], there have been many follow-up methods [1, 16]. However, several recent works have shown that simple baselines such as "number of parameters" and "FLOPS" are competitive with all existing ZC proxies across most settings, and that most ZC proxies do not generalize well across different benchmarks, thus requiring broader large-scale evaluations in order to assess their strengths [2, 25]. A recent landscape overview concluded that ZC proxies show great promise, but certain aspects are under-studied and their true

---

*Equal contribution. Work done while RT was part-time at Abacus.AI. Email to:
{krishnan, zelaa, fh}@cs.uni-freiburg.de, colin@abacus.ai, renbo.tu@mail.utoronto.ca,
safarim@informatik.uni-freiburg.de.

36th Conference on Neural Information Processing Systems (NeurIPS 2022) Track on Datasets and Benchmarks.

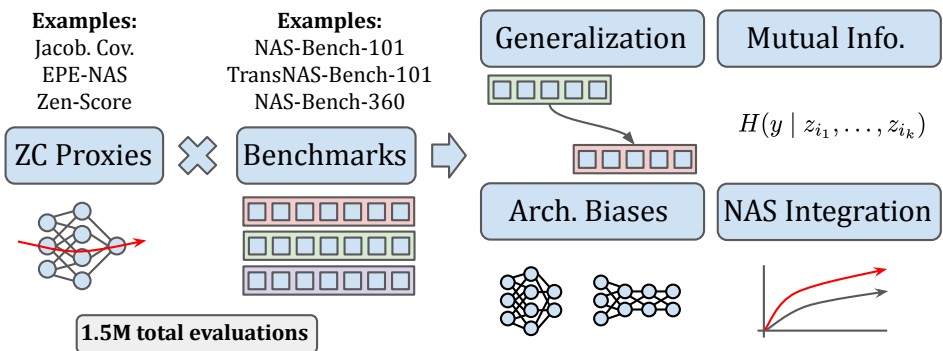

Figure 1: Overview of `NAS-Bench-Suite-Zero`. We implement and pre-compute 13 ZC proxies on 28 tasks in a unified framework, and then use this dataset to analyze the generalizability, complementary information, biases, and NAS integration of ZC proxies.

potential has not been realized thus far [45]. In particular, it is still largely unknown whether ZC proxies can be effectively combined, and how best to integrate ZC proxies into NAS algorithms.

In this work, we introduce `NAS-Bench-Suite-Zero`: a unified and extensible collection of 13 ZC proxies, accessible through a unified interface, which can be evaluated on a suite of 28 tasks through `NASLib` [30] (see Figure 1). In addition to the codebase itself, we release precomputed ZC proxy scores across all 13 ZC proxies and 28 tasks, which can be used to speed up ZC proxy experiments. Specifically, we show that the runtime of ZC proxy experiments such as NAS analyses and bias analyses are shortened by a factor of at least $10^3$ when using the precomputed ZC proxies in `NAS-Bench-Suite-Zero`. By providing a unified framework with ready-to-use scripts to run large-scale experiments, `NAS-Bench-Suite-Zero` eliminates the overhead for researchers to compare against many other methods and across all popular NAS benchmark search spaces, helping the community to rapidly increase the speed of research in this promising direction. Our benchmark suite was very recently used successfully in the Zero Cost NAS Competition at AutoML-Conf 2022. See Appendix F for more details. In Appendix B, we give detailed documentation, including a datasheet [10], license, author responsibility, code of conduct, and maintenance plan. We welcome contributions from the community and hope to grow the repository and benchmark suite as more ZC proxies and NAS benchmarks are released.

To demonstrate the usefulness of `NAS-Bench-Suite-Zero`, we run a large-scale analysis of ZC proxies: we give a thorough study of generalizability and biases, and we give the first information-theoretic analysis. Interestingly, based on the bias study, **we present a concrete method for improving the performance of a ZC proxy by reducing biases** (such as the tendency to favor larger architectures or architectures with more `conv` operations). This may have important consequences for the future design of ZC proxies. Furthermore, based on the information-theoretic analysis, we find that there is high information gain of the validation accuracy when conditioned on multiple ZC proxies, suggesting that ZC proxies do indeed compute substantial complementary information. Motivated by these findings, we incorporate all 13 proxies into the surrogate models used by NAS algorithms [44, 47], showing that the Spearman rank correlation of the surrogate predictions can *increase by up to 42%*. We show that this results in improved performance for two predictor-based NAS algorithms: BANANAS [47] and NPENAS [44].

**Our contributions.** We summarize our main contributions below.

- We release `NAS-Bench-Suite-Zero`, a collection of benchmarks and ZC proxies that unifies and accelerates research on ZC proxies – a promising new sub-field of NAS – by enabling orders-of-magnitude faster evaluations on a large suite of diverse benchmarks.
- We run a large-scale analysis of 13 ZC proxies across 28 different combinations of search spaces and tasks by studying the generalizability, bias, and mutual information among ZC proxies.
- Motivated by our analysis, we present a procedure to improve the performance of ZC proxies by reducing biases, and we show that the complementary information of ZC proxies can significantly improve the predictive power of surrogate models commonly used for NAS.

Table 1: List of ZC proxies in `NAS-Bench-Suite-Zero`. Note that "neuron-wise" denotes whether the total score is a sum of individual weights.

| Name | Data-dependent | Neuron-wise | Type | In `NAS-Bench-Suite-Zero` |
|------|:---:|:---:|:---:|:---:|
| `epe-nas` [21] | ✓ | ✗ | Jacobian | ✓ |
| `fisher` [42] | ✓ | ✓ | Pruning-at-init | ✓ |
| `flops` [25] | ✓ | ✓ | Baseline | ✓ |
| `grad-norm` [1] | ✓ | ✓ | Pruning-at-init | ✓ |
| `grasp` [43] | ✓ | ✓ | Pruning-at-init | ✓ |
| `l2-norm` [1] | ✗ | ✗ | Baseline | ✓ |
| `jacov` [23] | ✓ | ✗ | Jacobian | ✓ |
| `nwot` [23] | ✓ | ✗ | Jacobian | ✓ |
| `params` [25] | ✗ | ✓ | Baseline | ✓ |
| `plain` [1] | ✓ | ✓ | Baseline | ✓ |
| `snip` [14] | ✓ | ✓ | Pruning-at-init | ✓ |
| `synflow` [39] | ✗ | ✓ | Pruning-at-init | ✓ |
| `zen-score` [16] | ✗ | ✗ | Piece. Lin. | ✓ |

## 2 Background and Related Work

Given a dataset and a *search space* – a large set of neural architectures – NAS seeks to find the architecture with the highest validation accuracy (or the best application-specific trade-off among accuracy, latency, size, and so on) on the dataset. NAS has been studied since the late 1980s [24, 40] and has seen a resurgence in the last few years [18, 58], with over 1000 papers on NAS in the last two years alone. For a survey of the different techniques used for NAS, see [9, 49].

Many NAS methods make use of performance prediction. A *performance prediction* method is any function which predicts the (relative) performance of architectures, without fully training the architectures [48]. BRP-NAS [8], BONAS [34], and BANANAS [47] are all examples of NAS methods that make use of performance prediction. While performance prediction speeds up NAS algorithms by avoiding fully training neural networks, many still require non-trivial computation time. On the other hand, a recently-proposed line of techniques, *zero-cost proxies* (ZC proxies) require just a single forward pass through the network, often taking just five seconds [23].

**Zero-cost proxies.** The original ZC proxy estimated the separability of the minibatch of data into different linear regions of the output space [23]. Many other ZC proxies have been proposed since then, including data-independent ZC proxies [1, 15, 16, 39], ZC proxies inspired by pruning-at-initialization techniques [1, 14, 39, 43], and ZC proxies inspired by neural tangent kernels [4, 35]. See Table 1 for a full list of the ZC proxies we use in this paper. We describe theoretical ZC proxy results in Appendix C.1.

**Search spaces and tasks.** In our experiments, we make use of several different NAS benchmark search spaces and tasks. NAS-Bench-101 [54] is a popular cell-based search space for NAS research. It consists of 423 624 architectures trained on CIFAR-10. The cell-based search space is designed to model ResNet-like and Inception-like cells [12, 37]. NAS-Bench-201 [6] is a cell-based search space consisting of 15 625 architectures (6 466 non-isomorphic) trained on CIFAR-10, CIFAR-100, and ImageNet16-120. NAS-Bench-301 [56] is a surrogate NAS benchmark for the DARTS search space [19]. The search space consists of normal cell and reduction cells, with $10^{18}$ total architectures. TransNAS-Bench-101 [7] is a NAS benchmark consisting of two different search spaces: a "micro" (cell-based) search space of size 4 096, and a macro search space of size 3 256. The architectures are trained on seven different tasks from the Taskonomy dataset [55]. NAS-Bench-Suite [22] collects these search spaces and tasks within the unified framework of `NASLib` [30]. In this work, we extend this collection by adding two datasets from NAS-Bench-360 [41], SVHN, and four datasets from Taskonomy. NAS-Bench-360 is a collection of diverse tasks that are ready-to-use for NAS research.

**Large-scale studies of ZC proxies.** A few recent works [2, 25, 45, 48] investigated the performance of ZC proxies in ranking architectures over different NAS benchmarks, showing that the relative performance highly depends on the search space, but none study more than 12 total tasks, and none make the ZC proxy values publicly available. Two predictor-based NAS methods have recently been introduced: OMNI [48] and ProxyBO [33]. However, OMNI only uses a single ZC proxy, and

Table 2: Overview of ZC proxy evaluations in `NAS-Bench-Suite-Zero`. * Note that EPE-NAS is only defined for classification tasks [21].

| Search space | Tasks | Num. ZC proxies | Num. architectures | Total ZC proxy evaluations |
|---|---|---|---|---|
| NAS-Bench-101 | 1 | 13 | 10 000 | 130 000 |
| NAS-Bench-201 | 3 | 13 | 15 625 | 609 375 |
| NAS-Bench-301 | 1 | 13 | 11 221 | 145 873 |
| TransNAS-Bench-101-Micro | 7 | 12* | 3 256 | 273 504 |
| TransNAS-Bench-101-Macro | 7 | 12* | 4 096 | 344 064 |
| Add'l. 201, 301, TNB-Micro | 9 | 13 | 600 | 23400 |
| **Total** | **28** | **13** | **44 798** | **1 526 216** |

while ProxyBO uses three, the algorithm dynamically chooses one in each iteration (so individual predictions are made using a single ZC proxy at a time). Recently, NAS-Bench-Zero was introduced [2], a new benchmark based on popular computer vision models ResNet [12] and MobileNetV2 [31], which includes 10 ZC proxies. However, the NAS-Bench-Zero dataset is currently not publicly available. For more related work details, see Appendix C.

Only two prior works combine the information of multiple ZC proxies together in architecture predictions [1, 2] and both only use the *voting* strategy to combine at most four ZC proxies. Our work is the first to publicly release ZC proxy values, combine ZC proxies in a nontrivial way, and exploit the complementary information of 13 ZC proxies simultaneously.

## 3 Overview of NAS-Bench-Suite-Zero

In this section, we give an overview of the `NAS-Bench-Suite-Zero` codebase and dataset, which allows researchers to quickly develop ZC proxies, compare against existing ZC proxies across diverse datasets, and integrate them into NAS algorithms, as shown in Sections 4 and 5.

We implement all ZC proxies from Table 1 in the same codebase (`NASLib` [30]). For all ZC proxies, we use the default implementation from the original work. While this list covers 13 ZC proxies, the majority of ZC proxies released to date, we did not yet include a few other ZC proxies, for example, due to requiring a trained supernetwork to make evaluations [4, 35] (therefore needing to implement a supernetwork on 28 benchmarks), implementation in TensorFlow rather than PyTorch [26], or unreleased code. Our modular framework easily allows additional ZC proxies to be added to `NAS-Bench-Suite-Zero` in the future.

To build `NAS-Bench-Suite-Zero`, we extend the collection of `NASLib`'s publicly available benchmarks, known as NAS-Bench-Suite [22]. This allows us to evaluate and fairly compare all ZC proxies in the same framework without confounding factors stemming from different implementations, software versions or training pipelines. Specifically, for the search spaces and tasks, we use NAS-Bench-101 (CIFAR-10), NAS-Bench-201 (CIFAR-10, CIFAR-100, and ImageNet16-120), NAS-Bench-301 (CIFAR-10), and TransNAS-Bench-101 Micro and Macro (Jigsaw, Object Classification, Scene Classification, Autoencoder) from NAS-Bench-Suite. We add the remaining tasks from TransNAS-Bench-101 (Room Layout, Surface Normal, Semantic Segmentation), and three tasks each for NAS-Bench-201, NAS-Bench-301, and TransNAS-Bench-101-Micro: Spherical-CIFAR-100, NinaPro, and SVHN. This yields a total of 28 benchmarks in our analysis. For all NAS-Bench-201 and TransNAS-Bench-101 tasks, we evaluate all ZC proxy values and the respective runtimes, for all architectures. For NAS-Bench-301, we evaluate on all 11 221 randomly sampled architectures from the NAS-Bench-301 dataset, due to the computational infeasibility of exhaustively evaluating the full set of $10^{18}$ architectures. Similarly, we evaluate 10 000 architectures from NAS-Bench-101. Finally, for Spherical-CIFAR-100, NinaPro, and SVHN, we evaluate 200 architectures per search space, since only 200 architectures are fully trained for each of these tasks. See Table 2.

We run all ZC proxies from Table 1 on Intel Xeon Gold 6242 CPUs and save their evaluations in order to create a queryable table with these pre-computed values. We use a batch size of 64 for all ZC proxy evaluations, except for the case of TransNAS-Bench-101: due to the extreme memory usage of the Taskonomy tasks ($> 30$GB memory), we used a batch size of 32. The total computation time for all 1.5M evaluations was 1100 CPU hours.

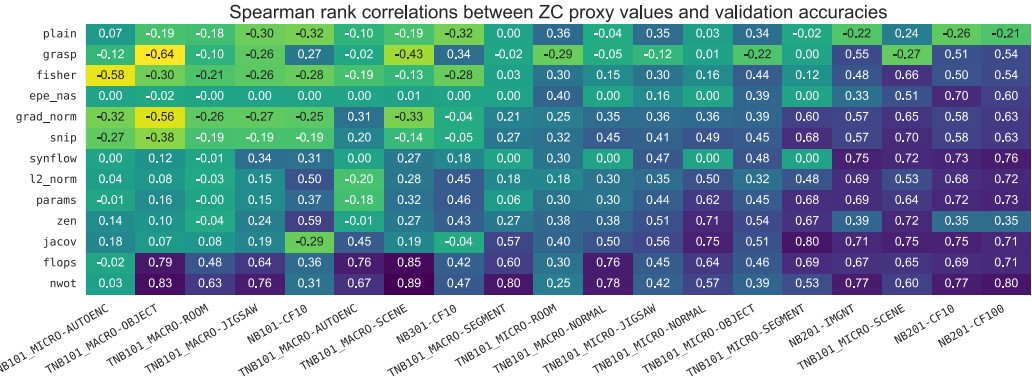

Figure 2: Spearman rank correlation coefficient between ZC proxy values and validation accuracies, for each ZC proxy and benchmark. The rows and columns are ordered based on the mean scores across columns and rows, respectively.

**Speedups and recommended usage.** The average time to compute a ZC proxy across all tasks is 2.6 seconds, and the maximum time (computing `grasp` on TNB-Macro Autoencoder) is 205 seconds, compared to $10^{-5}$ seconds when instead querying the `NAS-Bench-Suite-Zero`API.

When researchers evaluate ZC proxy-based NAS algorithms using queryable NAS benchmarks, the bottleneck is often (ironically) the ZC proxy evaluations. For example, for OMNI [48] or ProxyBO [33] running for 100 iterations and 100 candidates per iteration, the total evaluation time is roughly 9 hours, yet they can be run on `NAS-Bench-Suite-Zero` in under one minute. Across all experiments done in this paper (mutual information study, bias study, NAS study, etc.), we calculate that using `NAS-Bench-Suite-Zero` decreases the computation time by at least three orders of magnitude. See Appendix D.4 for more details.

Since `NAS-Bench-Suite-Zero` reduces the runtime of experiments by at least three orders of magnitude (on queryable NAS benchmarks), we recommend researchers take advantage of `NAS-Bench-Suite-Zero` to *(i)* run hundreds of trials of ZC proxy-based NAS algorithms, to reach statistically significant conclusions, *(ii)* run extensive ablation studies, including the type and usage of ZC proxies, and *(iii)* increase the total number of ZC proxies evaluated in the NAS algorithm. Finally, when using `NAS-Bench-Suite-Zero`, researchers should report the real-world time NAS algorithms would take, by adding the time to run each ZC proxy evaluation (which can be queried in `NAS-Bench-Suite-Zero`) to the total runtime of the NAS algorithm.

## 4 Generalizability, Mutual Information, and Bias of ZC Proxies

In this section, we use `NAS-Bench-Suite-Zero` to study concrete research questions relating to the generalizability, complementary information, and bias of ZC proxies.

### 4.1 RQ 1: How well do ZC proxies generalize across different benchmarks?

In Figure 2, for each ZC proxy and each benchmark, we compute the Spearman rank correlation between the ZC proxy values and the validation accuracies over a set of 1000 randomly drawn architectures (see Appendix D for the full results on all benchmarks). Out of all the ZC proxies, `nwot` and `flops` have the highest rank correlations across all benchmarks. On some of the benchmarks, such as TransNAS-Bench-101-Micro Autoencoder and Room Layout, all of the ZC proxies exhibit poor performance on average, while on the widely used NAS-Bench-201 benchmarks, almost all of them perform well. Several methods, such as `snip` and `grasp`, perform well on the NAS-Bench-201 tasks, but on average are outperformed by `params` and `flops` on the other benchmarks.

Although no ZC proxy performs consistently across all benchmarks, we may ask a related question: is the performance of all ZC proxies across benchmarks correlated enough to capture similarities among benchmarks? In other words, can we use ZC proxies as a tool to assess the similarities among tasks. This is particularly important in meta-learning or transfer learning, where a meta-algorithm aims to learn and transfer knowledge across a set of similar tasks. To answer this question, we

compute the Pearson correlation of the ZC proxy scores on each pair of benchmarks. See Figure 3. As expected, benchmarks that are based on the same or similar search spaces are highly correlated with respect to the ZC proxy scores. For example, we see clusters of high correlation for the Trans-NAS-Bench-101-Macro benchmarks, and the NAS-Bench-201 benchmarks.

**Answer to RQ 1:** *Only a few ZC proxies generalize well across most benchmarks and tasks. However, ZC proxies can be used to assess similarities across benchmarks.* This suggests the potential future direction of incorporating them as task features in a meta-learning setting [20].

### 4.2 RQ 2: Are ZC proxies complementary with respect to explaining validation accuracy?

While Figure 2 shows the performance of each individual ZC proxy, now we consider the combined performance of multiple ZC proxies. If ZC proxies measure different characteristics of architectures, then a NAS algorithm can exploit their complementary information in order to yield improved results. While prior work [25, 45] computes the correlation among pairs of ZC proxies, [2] our true goal is to assess the complementary information of ZC proxies *with respect to explaining the ground-truth validation accuracy*. Furthermore, we wish to measure the complementary information of more than just two ZC proxies at a time. For this, we turn to information theoretic measures: by treating the validation accuracy and ZC proxy values as random variables, we can measure the entropy

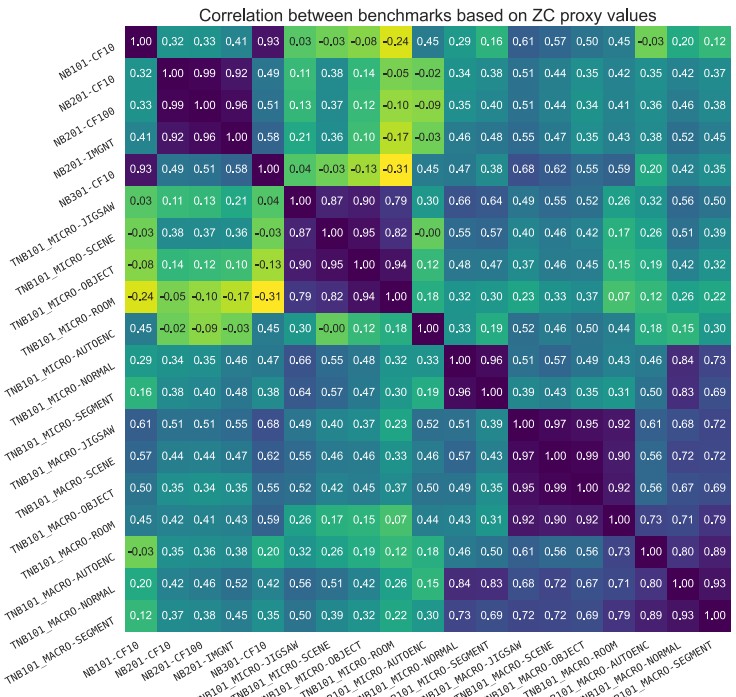

Figure 3: Pearson correlation coefficient between ZC proxy scores on pairs of benchmarks. The entries in the plot are ordered based on the mean score across each row and column.

of the validation accuracy conditioned on one or more ZC proxies, which intuitively tells us the information that one or more ZC proxies reveal about the validation accuracy.

Formally, given a search space $S$, let $\mathcal{Y}$ denote the uniform distribution of validation accuracies over the search space, and let $y$ denote a random sample from $\mathcal{Y}$. Similarly, for a ZC proxy $i$ from 1 to 13, let $\mathcal{Z}_i$ denote the uniform distribution of the ZC proxy values, and let $z_i$ denote a random sample from $\mathcal{Z}_i$. Let $H(\cdot)$ denote the entropy function. For all pairs $z_i, z_j$ of ZC proxies, we compute the conditional entropy $H(y \mid z_i, z_j)$, as well as the information gain $H(y \mid z_i) - H(y \mid z_i, z_j)$. See Figure 4. The entropy computations are based on 1000 randomly sampled architectures, using 24-bin histograms for density smoothing (see Appendix D for more details). We see that synflow and plain together give the most information about the ground truth validation accuracies, due to their substantial complementary information.

Now we can ask the same question for $k$ tuples of ZC proxies. Given an ordered list of $k$ ZC proxies $z_{i_1}, z_{i_2}, \dots z_{i_k}$, we define the information gain of $z_{i_k}$ conditioned on $y$ as follows:

$$\mathbf{IG}(z_{i_k}) := H(y \mid z_{i_1}, \dots, z_{i_{k-1}}) - H(y \mid z_{i_1}, \dots, z_{i_k}). \tag{1}$$

Intuitively, **IG** computes the marginal information we learn about $y$ when $z_{i_k}$ is revealed, assuming we already knew the values of $z_{i_1}, \dots, z_{i_{k-1}}$. We compare the conditional entropy vs. number of

---

[2]For completeness, we re-run that experiment and include the results in Appendix D.

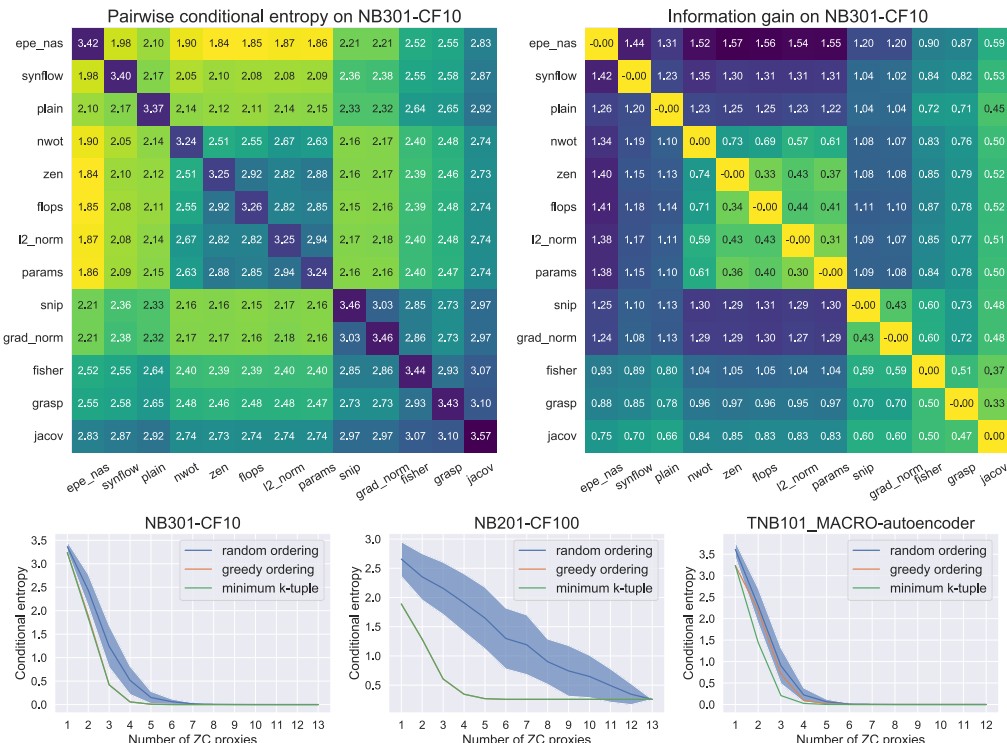

Figure 4: Given a ZC proxy pair $(i, j)$, we compute the conditional entropy $H(y \mid z_i, z_j)$ (top left), and information gain $H(y \mid z_i) - H(y \mid z_i, z_j)$ (top right). Conditional entropy $H(y \mid z_{i_1}, \ldots, z_{i_k})$ vs. $k$, where the ordering $z_{i_1}, \ldots, z_{i_k}$ is selected using three different strategies. The minimum $k$-tuple and greedy ordering significantly overlap in the first two figures (bottom).

ZC proxies for three different orderings of the ZC proxies. The first is a random ordering (averaged over 100 random trials), which tells us the average information gain when iteratively adding more ZC proxies. The second is a greedy ordering, computed by iteratively selecting the ZC proxy that maximizes $\mathbf{IG}(z_{i_k})$, for $k$ from 1 to 13. The final plot exhaustively searches through $\binom{13}{k}$ sets to find the $k$ proxies which minimize $H(y \mid z_{i_1}, \ldots z_{i_k})$, for $k$ from 1 to 13 (note that this may not define a valid ordering). See Figure 4, and Appendix D for the complete results. We see that there is very substantial information gain when iteratively adding ZC proxies, even if the ZC proxies are randomly chosen. Optimizing the order of adding ZC proxies yields much higher $\mathbf{IG}$ in certain benchmarks (e.g., NB201-CF100), and a greedy approach is shown to be not far from the optimum.

**Answer to RQ 2:** *In some benchmarks, we see substantial complementary information among ZC proxies. However, the degree of complementary information depends heavily on the NAS benchmark at hand.* This suggests that we cannot always expect ZC proxies to yield complementary information, but a machine learning model might be able to identify useful combinations of ZC proxies.

### 4.3 RQ 3: Do ZC proxies contain biases, such as a bias toward certain operations or sizes, and can we mitigate these biases?

Identifying biases in ZC proxies can help explain weaknesses and facilitate the development of higher-performing ZC proxies. We define bias metrics and study ZC proxy scores for thousands of architectures for their correlation with biases. This systematic approach yields generalizable conclusions and avoids the noise from assessing singular architectures. We consider the following biases: **conv:pool** (the numerical advantage of convolution to pooling operations in the cell), **cell size** (the number of non-zero operations in the cell), **num. skip connections**, and **num. parameters**.

For each search space, ZC proxy, and bias, we compute the Pearson correlation coefficient between the ZC proxy values and the bias values. We consider all 44K architectures referenced in Table 2. See Table 3 and Appendix D for the full results. We find that many ZC proxies exhibit biases to

Table 3: Pearson correlation coefficients between predictors and bias metrics (in bold) on different datasets. For example, for **Cell size** on NB201-CF100, `snip` has a correlation of -0.04 (indicating very little bias), while `synflow` has a correlation of 0.57 (meaning it favors larger architectures).

| Name | Conv:pool | | Cell size | | Num. skip connections | | Num. parameters | |
|---|---|---|---|---|---|---|---|---|
| | NB201-CF10 | NB301-CF10 | NB201-CF100 | NB201-IM | NB301-CF10 | NB201-CF100 | NB101-CF10 | NB301-CF10 |
| epe-nas | 0.05 | -0.02 | 0.35 | 0.35 | 0.01 | 0.09 | -0.02 | -0.01 |
| fisher | 0.05 | 0.01 | -0.03 | -0.05 | -0.15 | -0.03 | 0.11 | 0.17 |
| flops | 0.59 | 0.70 | 0.30 | 0.30 | **-0.35** | -0.30 | **1.00** | 0.99 |
| grad-norm | 0.35 | 0.27 | -0.04 | -0.05 | -0.26 | -0.26 | 0.30 | 0.51 |
| grasp | 0.01 | 0.28 | -0.01 | 0.01 | 0.03 | 0.00 | -0.03 | 0.24 |
| l2-norm | **0.87** | 0.76 | 0.41 | 0.41 | -0.33 | **-0.41** | 0.62 | 0.99 |
| jacov | 0.05 | -0.11 | 0.35 | 0.35 | 0.08 | 0.09 | -0.18 | -0.10 |
| nwot | 0.06 | **0.78** | 0.28 | 0.28 | -0.21 | 0.06 | 0.74 | 0.95 |
| params | 0.61 | **0.78** | 0.29 | 0.29 | -0.32 | -0.29 | **1.00** | **1.00** |
| plain | -0.33 | -0.45 | 0.14 | 0.14 | 0.02 | 0.02 | 0.03 | -0.45 |
| snip | 0.37 | 0.27 | -0.04 | -0.04 | -0.28 | -0.28 | 0.44 | 0.50 |
| synflow | 0.53 | 0.41 | **0.57** | **0.58** | -0.20 | -0.14 | 0.57 | 0.62 |
| zen-score | 0.05 | 0.75 | 0.35 | 0.35 | -0.33 | 0.09 | 0.68 | 0.99 |
| val-acc | 0.36 | 0.45 | 0.35 | 0.43 | 0.13 | -0.06 | 0.09 | 0.47 |

various degrees. Interestingly, some biases are consistent across search spaces, while others are not. For example, `l2-norm` has a conv:pool bias on both NB201-C10 and NB301-C10, while `nwot` has a strong conv:pool bias on NB301-C10 and almost no bias on NB201-C10. While validation accuracy does not correlate with number of skip connections, most ZC proxies in the benchmark exhibit a negative bias towards this metric.

Next, we present a procedure for removing these biases. For this study, we use ZC proxies that had large biases in Table 3, and we attempt to answer the following questions: *(1)* can we remove these biases, and *(2)* if we can remove the biases, does the performance of ZC proxies improve?

Given a search space of architectures $A$, let $f : A \rightarrow \mathbb{R}$ denote a ZC proxy (a function that takes as input an architecture, and outputs a real number). Furthermore, let $b : A \rightarrow \mathbb{R}$ denote a bias measure such as "cell size". Recall that Table 3 showed that the correlation between a ZC proxy $f$ and a bias measure $b$ may be high. For example, the correlation between `synflow` and "cell size" is high, which means using `synflow` would favor larger architectures. To reduce bias, we use a simple heuristic:

$$f'(a) = f(a) \cdot \frac{1}{b(a) + C}. \tag{2}$$

In this expression, $C$ is a constant that we can tune. In deciding on a strategy to tune $C$, we make two observations. First, for most bias measures, the bias of `val_acc` is not zero, which means completely de-biasing ZC proxies could hurt performance. Second, depending on the application, we may want to fully remove the bias of a ZC proxy, or else remove bias only insofar as it improves performance.

Therefore, we test three different strategies to tune $C$ by brute force: *(1)* "minimize", to minimize bias, *(2)* "equalize", to match the bias with the bias of `val_acc`, and *(3)* "performance", to optimize the performance (Pearson correlation). See Table 4 for the results.

We find that using the "performance" strategy, we are able to increase the performance of ZC proxies by reducing their bias. Furthermore, the "equalize" strategy sometimes provide good results on par with the "performance" strategy. This suggests a good bias mitigation strategy when we do not know the ground truth but have information on how the ground truth correlations with bias. This may have important consequences for the future design of ZC proxies.

**Answer to RQ 3:** *Many ZC proxies do exhibit different types of biases to various degrees, but the biases can be mitigated, thereby improving performance.*

## 5 Integration into NAS

The findings in Section 4.2 showed that ZC proxies contain substantial complementary information, conditioned on the ground-truth validation accuracies. However, no prior work has combined more than four ZC proxies, or used a combination strategy other than a simple vote. In this section, we combine and integrate all 13 ZC proxies into predictor-based NAS algorithms by adding the ZC proxies directly as features into the surrogate (predictor) models.

Table 4: Bias mitigation strategies tested on the ZC proxies with the most biases. We test three different strategies by tuning $C$ from Equation 2 for different objectives: minimize (tune $C$ to minimize bias), equalize (tune $C$ to match ground truth's correlation with bias metric), and performance (tune $C$ to maximize correlation with ground truth). Bias and performance are Pearson correlation coefficients of the proxy score with the bias metric and with the ground truth accuracy, respectively. $C$ is searched between -10 and 1000.

| ZC proxy | dataset | bias metric | original bias | original perf. | new bias | new perf. | strategy |
|---|---|---|---|---|---|---|---|
| l2-norm | NB201-CF10 | conv:pool | 0.87 | 0.42 | 0.00 | 0.10 | minimize |
|  |  |  |  |  | 0.37 | 0.11 | equalize |
|  |  |  |  |  | 0.70 | 0.44 | performance |
| nwot | NB301-CF10 | conv:pool | 0.78 | 0.49 | 0.00 | 0.03 | minimize |
|  |  |  |  |  | 0.29 | 0.14 | equalize |
|  |  |  |  |  | 0.78 | 0.49 | performance |
| synflow | NB201-CF100 | cell size | 0.57 | 0.68 | 0.01 | 0.64 | minimize |
|  |  |  |  |  | 0.35 | 0.71 | equalize |
|  |  |  |  |  | 0.35 | 0.71 | performance |
| synflow | NB201-IM | cell size | 0.58 | 0.76 | 0.01 | 0.62 | minimize |
|  |  |  |  |  | 0.43 | 0.76 | equalize |
|  |  |  |  |  | 0.46 | 0.76 | performance |
| flops | NB301-CF10 | num. skip | -0.35 | 0.43 | -0.01 | 0.06 | minimize |
|  |  |  |  |  | 0.12 | -0.05 | equalize |
|  |  |  |  |  | -0.35 | 0.43 | performance |

We run experiments on two common predictor-based NAS algorithms: BANANAS, based on Bayesian optimization [47], and NPENAS, based on evolution [44]. Both algorithms use a model-based performance predictor: a model that takes in an architecture encoding as features (e.g., the adjacency matrix encoding [46]), and outputs a prediction of that architecture's validation accuracy. The model is retrained throughout the search algorithm, as more and more architectures are fully trained. Recent work has shown that boosted trees such as XGBoost achieve strong performance in NAS [48, 56].

**Experimental setup.** For both algorithms, we use the `NASLib` implementation [30] and default parameters reported in prior work [48]. First, we assess the standalone performance of XGBoost when ZC proxies are added as features in addition to the architecture encoding, by randomly sampling 100 training architectures and 1000 disjoint test architectures, and computing the Spearman rank correlation coefficient between the set of predicted validation accuracies and the ground-truth accuracies. On NAS-Bench-201 CIFAR-100, averaged over 100 trials, the Spearman rank correlation ($\pm$ std. dev.) improves from $0.640 \pm 0.0420$ to $\mathbf{0.908 \pm 0.012}$ with the addition of ZC proxies, representing an *improvement of 41.7%*. Even more surprisingly, using the ZC proxies alone as features without the architecture, results in a Spearman rank correlation of $\mathbf{0.907 \pm 0.013}$, implying that the ZC proxies subsume nearly all information contained in the architecture encoding itself. We present the full results in Appendix E. These results show that an ensemble of ZC proxies can substantially increase the performance of model-based predictors.

Similar to the previous experiment, we run both NAS algorithms three different ways: using only the encoding, only the ZC proxies, and both, as features of the predictor. Each algorithm is given 200 architecture evaluations, and we plot performance over time, averaged over 400 trials. See Figure 5 for the results of BANANAS, and Appendix E for the full results. We find that the ZC proxies give the NAS algorithms a boost in performance, especially in the early stages of the search.

## 6  Conclusions, Limitations, and Broader Impact

In this work, we created `NAS-Bench-Suite-Zero`: an extensible collection of 13 ZC proxies (covering the majority that currently exist), accessible through a unified interface, which can be evaluated on a suite of 28 NAS benchmark tasks. In addition to the codebase, we release precomputed ZC proxy scores across all 13 ZC proxies and 28 tasks, giving 1.5 million total ZC proxy evaluations. This dataset can be used to speed up ZC proxy-based NAS experiments, e.g., from 9 hours to 4

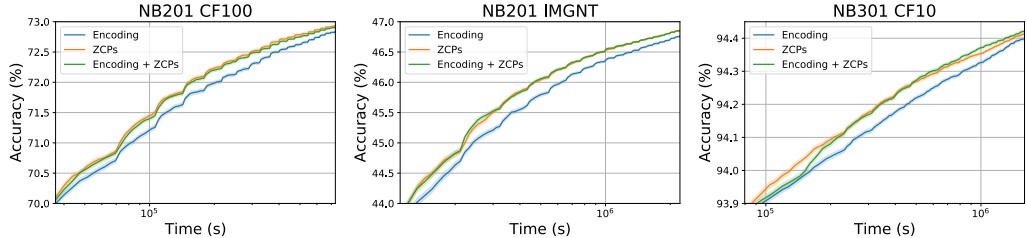

Figure 5: Performance of BANANAS with and without ZC proxies as additional features in the surrogate model. Each curve shows the mean and standard error across 400 trials.

minutes (see Section 3). Overall, NAS-Bench-Suite-Zero eliminates the overhead in ZC proxy research, with respect to comparing against different methods and across a diverse set of tasks.

To motivate the usefulness of NAS-Bench-Suite-Zero, we conducted a large-scale analysis of the generalizability, bias, and the first information-theoretic analysis of ZC proxies. Our empirical analysis showed substantial complementary information of ZC proxies conditioned on validation accuracy, motivating us to ensemble all 13 into predictor-based NAS algorithms. We show that using several ZC proxies together significantly improves the performance of the surrogate models used in NAS, as well as improving the NAS algorithms themselves.

**Limitations and future work.** Although our work makes substantial progress towards motivating and increasing the speed of ZC proxy research, there are still some limitations of our analysis. First, our work is limited to empirical analysis. However, we discuss existing theoretical results in Appendix C.1. Furthermore, there are some benchmarks on which we did not give a comprehensive evaluation. For example, on NAS-Bench-301, we only computed ZC proxies on $11\,000$ architectures, since the full space of $10^{18}$ architectures is computationally infeasible. In the future, a surrogate model [53, 56] could be trained to predict the performance of ZC proxies on the remaining architectures. Finally, there is very recent work on applying ZC proxies to one-shot NAS methods [52], which tested one ZC proxy at a time with one-shot models. Since our work motivates the ensembling of ZC proxies, an exciting problem for future work is to incorporate 13 ZC proxies into the one-shot framework.

**Broader impact.** The goal of our work is to make it faster and easier for researchers to run reproducible, generalizable ZC proxy experiments and to motivate further study on exploiting the complementary strengths of ZC proxies. By pre-computing ZC proxies across many benchmarks, researchers can run many trials of NAS experiments cheaply on a CPU, reducing the carbon footprint of the experiments [11, 27]. Due to the notoriously high GPU consumption of prior research in NAS [28, 58], this reduction in CO2 emissions is especially worthwhile. Furthermore, our hope is that our work will have a positive impact on the NAS and automated machine learning communities by showing which ZC proxies are useful in which settings, and showing how to most effectively combine ZC proxies to achieve the best predictive performance. By open-sourcing all of our code and datasets, AutoML researchers can use our library to further test and develop ZC proxies for NAS.

## Acknowledgments and Disclosure of Funding

This research was supported by the following sources: Robert Bosch GmbH is acknowledged for financial support; the German Federal Ministry of Education and Research (BMBF, grant Renormal-izedFlows 01IS19077C); TAILOR, a project funded by EU Horizon 2020 research and innovation programme under GA No 952215; the Deutsche Forschungsgemeinschaft (DFG, German Research Foundation) under grant number 417962828; the European Research Council (ERC) Consolidator Grant "Deep Learning 2.0" (grant no. 101045765). Funded by the European Union. Views and opinions expressed are however those of the author(s) only and do not necessarily reflect those of the European Union or the ERC. Neither the European Union nor the ERC can be held responsible for them.

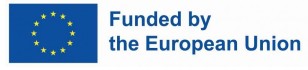

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
