## A  NAS Best Practices Checklist

We now describe how we addressed the individual points of the NAS best practice checklist [17].

1. **Best Practices for Releasing Code**

   For all experiments you report:

   (a) Did you release code for the training pipeline used to evaluate the final architectures? [Yes] Since we used NAS benchmarks, we did not evaluate the architectures ourselves. The code for the training pipelines of these benchmarks is publicly available.

   (b) Did you release code for the search space [Yes] Since we used NAS benchmarks, this is already publicly available.

   (c) Did you release the hyperparameters used for the final evaluation pipeline, as well as random seeds? [Yes] Since we used NAS benchmarks, the final evaluation pipeline is fixed. We released our code, including the seeds used.

   (d) Did you release code for your NAS method? [Yes] The code for our NAS method is available at `https://github.com/automl/naslib/tree/zerocost`.

   (e) Did you release hyperparameters for your NAS method, as well as random seeds? [Yes] The hyperparameters used are also available at the above link.

2. **Best practices for comparing NAS methods**

   (a) For all NAS methods you compare, did you use exactly the same NAS benchmark, including the same dataset (with the same training-test split), search space and code for training the architectures and hyperparameters for that code? [Yes] Since we used NAS benchmarks, the training details are fixed.

   (b) Did you control for confounding factors (different hardware, versions of DL libraries, different runtimes for the different methods)? [Yes] Since we used NAS Benchmarks, these details are fixed automatically.

   (c) Did you run ablation studies? [Yes] We included NAS experiments with only the encoding, only the ZC proxies, and the encoding with 13 ZC proxies.

   (d) Did you use the same evaluation protocol for the methods being compared? [Yes] We used NAS Benchmarks, which keep this fixed.

   (e) Did you compare performance over time? [Yes] Our experiments in Section 5 and Appendix E compare performance over time.

   (f) Did you compare to random search? [No] We used baselines that are better than random search: the original NAS algorithms without ZC proxies.

   (g) Did you perform multiple runs of your experiments and report seeds? [Yes] All of our experiments are averaged across many trials. The seeds are reported in our code files.

   (h) Did you use tabular or surrogate benchmarks for in-depth evaluations? [Yes] All of our experiments use queryable benchmarks.

3. **Best practices for reporting important details**

   (a) Did you report how you tuned hyperparameters, and what time and resources this required? [Yes] We used the default hyperparameters from the respective NAS algorithms and ZC proxies. Our addition of ZC proxies did not add any new hyperparameters.

   (b) Did you report the time for the entire end-to-end NAS method (rather than, e.g., only for the search phase)? [Yes] Our plots include the end-to-end time.

   (c) Did you report all the details of your experimental setup? [Yes] We included all the details in Section 5 and Appendix E.

## B  Dataset Documentation

Here, we give an overview of our dataset documentation. For the full details, including links to the dataset, usage, and tutorials, see `https://github.com/automl/NASLib/tree/zerocost`.

Table 5: Licenses for the datasets that we use.

| Dataset | License | URL |
|---|---|---|
| NAS-Bench-101 | Apache 2.0 | `https://github.com/google-research/nasbench` |
| NAS-Bench-201 | MIT | `https://github.com/D-X-Y/NAS-Bench-201` |
| NAS-Bench-301 | Apache 2.0 | `https://github.com/automl/nasbench301` |
| TransNAS-Bench-101 | MIT | `https://github.com/yawen-d/TransNASBench` |
| NAS-Bench-360 | MIT | `https://github.com/rtu715/NAS-Bench-360` |

## B.1 Author responsibility and license

We, the authors, bear all responsibility in case of violation of rights. The license of our dataset and repository is the **Apache License 2.0**. For more information, see `https://github.com/automl/NASLib/blob/Develop/LICENSE`.

In addition, we include the licenses of the datasets we used in Table 5.

## B.2 Maintenance plan

The data is available on GitHub at `https://github.com/automl/NASLib/tree/zerocost`. We plan to actively maintain the repository, and we also welcome contributions from the community. For more information, see `https://github.com/automl/NASLib/tree/zerocost`.

## B.3 Code of conduct

Our Code of Conduct is from the Contributor Covenant, version 2.0. See `https://www.contributor-covenant.org/version/2/0/code_of_conduct.html`. The policy is copied below.

> "We as members, contributors, and leaders pledge to make participation in our community a harassment-free experience for everyone, regardless of age, body size, visible or invisible disability, ethnicity, sex characteristics, gender identity and expression, level of experience, education, socio-economic status, nationality, personal appearance, race, caste, color, religion, or sexual identity and orientation."

## B.4 Datasheet

We include a datasheet [10] for `NAS-Bench-Suite-Zero`.

**Motivation For Datasheet Creation**

*Why was the datasheet created? (e.g., was there a specific task in mind? was there a specific gap that needed to be filled?)* The goal of our work is to make it easier and faster for researchers to run generalizable, reproducible ZC proxy experiments, and to motivate further study on exploiting the complementary strengths of ZC proxies. By pre-computing ZC proxies across many benchmarks, users can run many trials of NAS experiments cheaply on a CPU, reducing their carbon footprint [11, 27]. Since prior research in NAS has notoriously high GPU consumption [28, 58], this reduction in CO2 emissions is worthwhile.

*Has the dataset been used already? If so, where are the results so others can compare (e.g., links to published papers)?* The dataset has only been used in this paper. See Sections 4 and 5 and Appendix D and E.

*What (other) tasks could the dataset be used for? Since the dataset only contains values of ZC proxies on existing NAS benchmarks, we are not aware of any tasks this dataset can be used for, besides analyzing ZC proxies and speeding up ZC proxy-based NAS algorithms.

*Who funded the creation dataset? This dataset was created by researchers at the University of Freiburg, Abacus.AI, the University of Toronto, and the Bosch Center for Artificial Intelligence. Funding for the dataset computation itself is from the University of Freiburg.

*Any other comment? None.

**Datasheet Composition**

*What are the instances?(that is, examples; e.g., documents, images, people, countries) Are there multiple types of instances? (e.g., movies, users, ratings; people, interactions between them; nodes, edges) For each NAS benchmark, each instance is a tuple of an architecture hash, the name of a ZC proxy, and the value and runtime of the ZC proxy evaluated on that architecture.

*How many instances are there in total (of each type, if appropriate)? See Table 2 for a full breakdown of the number of instances for each NAS benchmark.

*What data does each instance consist of ? "Raw" data (e.g., unprocessed text or images)? Features/attributes? Is there a label/target associated with instances? If the instances related to people, are subpopulations identified (e.g., by age, gender, etc.) and what is their distribution? Each instance is a tuple of an architecture hash, the name of a ZC proxy, and the value and runtime of the ZC proxy evaluated on that architecture. These will most-often be used to speed up NAS experiments or run analysis on ZC proxies, in which case they are not used as features/labels.

*Is any information missing from individual instances? If so, please provide a description, explaining why this information is missing (e.g., because it was unavailable). This does not include intentionally removed information, but might include, e.g., redacted text. There is no missing information from individual instances.

*Are relationships between individual instances made explicit (e.g., users' movie ratings, social network links)? If so, please describe how these relationships are made explicit. There are no relationships between individual instances.

*Does the dataset contain all possible instances or is it a sample (not necessarily random) of instances from a larger set? If the dataset is a sample, then what is the larger set? Is the sample representative of the larger set (e.g., geographic coverage)? If so, please describe how this representativeness was validated/verified. If it is not representative of the larger set, please describe why not (e.g., to cover a more diverse range of instances, because instances were withheld or unavailable). NAS-Bench-201 and TransNAS-Bench-101-Micro and Macro contain all possible instances. NAS-Bench-101, NAS-Bench-301, and the additional architectures evaluated on spherical-cifar, SVHN, and NinaPro are samples. All samples are drawn uniformly at random from the respective search space. This is ensured because the code used to draw architectures uniformly at random is from the respective original repositories that introduced the NAS benchmarks.

 The main usage of this dataset is to speed up NAS experiments, for which there are no data splits. For experiments involving architecture prediction (such as the standalone predictor experiments in Section 5, we do not give recommended data splits but instead recommended running at least 100 trials, where each trial randomly samples train and (disjoint) test sets.

 There are no known errors, sources of noise, or redundancies.

 The dataset does rely on the code from the respective existing NAS benchmarks to reconstruct the architecture itself from the hash provided in our dataset. Furthermore, a user will often want access to the validation accuracies of the architectures in our dataset, which also comes from the existing NAS benchmarks. Since these NAS benchmarks serve similar goals as our dataset (to accelerate and simplify research in NAS) and are hosted similarly to ours (on Google Drive and GitHub), we are confident that these benchmarks will exist and remain constant over time. In some cases, we have also created our own versions of the NAS benchmarks, so all of the data can be downloaded at one time. Licenses and links are described in Table 5.

 None.

**Collection Process**

 The data was created with a software program (available at `https://github.com/automl/NASLib/tree/zerocost`). The ZC proxy code were taken from their original repositories. All ZC proxies from Table 1 were run on an Intel Xeon Gold 6242 CPU, using a batch size of 64, except for the case of TransNAS-Bench-101: due to the extreme memory usage of the Taskonomy tasks ($> 30$GB memory), we used a batch size of 32.

 As described, all data was created with a publicly available software program.

 As described earlier, the sampling was done uniformly at random.

*Who was involved in the data collection process (e.g., students, crowdworkers, contractors) and how were they compensated (e.g., how much were crowdworkers paid)? The data collection process (e.g., running the code) was done by the authors of this work.

*Over what timeframe was the data collected? Does this timeframe match the creation timeframe of the data associated with the instances (e.g., recent crawl of old news articles)? If not, please describe the timeframe in which the data associated with the instances was created. The total computation time for all 1.5M evaluations was 1100 CPU hours on Intel Xeon Gold 6242 CPUs (using up to 20 CPUs and 150 cores in parallel). The timeframe was May 15, 2022 to June 1, 2022.

**Data Preprocessing**

*Was any preprocessing/cleaning/labeling of the data done (e.g., discretization or bucketing, tokenization, part-of-speech tagging, SIFT feature extraction, removal of instances, processing of missing values)? If so, please provide a description. If not, you may skip the remainder of the questions in this section. There was no preprocessing that needed to be done.

*Does this dataset collection/processing procedure achieve the motivation for creating the dataset stated in the first section of this datasheet? If not, what are the limitations? Yes, the dataset collection procedure achieves our motivation. See Table 8 for a list of the speedups in NAS experiments achieved when using our dataset.

*Any other comments None.

**Dataset Distribution**

*How will the dataset be distributed? (e.g., tarball on website, API, GitHub; does the data have a DOI and is it archived redundantly?) The dataset is on Google Drive, with a DOI.

*When will the dataset be released/first distributed? What license (if any) is it distributed under? The dataset is public as of June 8, 2022, distributed under the Apache License 2.0.

*Are there any copyrights on the data? There are no copyrights on the data.

*Are there any fees or access/export restrictions? There are no fees or restrictions.

*Any other comments? None.

**Dataset Maintenance**

*Who is supporting/hosting/maintaining the dataset? The authors of this work are supporting/hosting/maintaining the dataset.

*Will the dataset be updated? If so, how often and by whom?* If new NAS benchmarks are created in the NAS research community, the authors of this work may update `NAS-Bench-Suite-Zero` to include ZC proxy values for the new benchmarks. Similarly, if new ZC proxies are relased, the authors may update `NAS-Bench-Suite-Zero` to include the new ZC proxies.

*How will updates be communicated? (e.g., mailing list, GitHub)* Updates will be communicated on the GitHub README of this project.

*If the dataset becomes obsolete how will this be communicated?* If the dataset becomes obsolete, it will be communicated on the GitHub README of this project.

*If others want to extend/augment/build on this dataset, is there a mechanism for them to do so? If so, is there a process for tracking/assessing the quality of those contributions. What is the process for communicating/distributing these contributions to users?* Others can create a pull request or raise an issue on GitHub with possible extensions/augmentations to our dataset, which will be approved in a case-by-case basis. For example, an author of a new ZC proxy may create a PR in our codebase with the new ZC proxy, and then we will evaluate the ZC proxy on all architectures in `NAS-Bench-Suite-Zero` and update the dataset. These updates will again be communicated on the GitHub README.

**Legal and Ethical Considerations**

*Were any ethical review processes conducted (e.g., by an institutional review board)? If so, please provide a description of these review processes, including the outcomes, as well as a link or other access point to any supporting documentation.* There was no ethical review process. We note that our dataset was created by simply by running ZC proxy computations on architectures of existing NAS benchmarks, in some cases using publicly available, licensed datasets such as CIFAR-10 or CIFAR-100.

*Does the dataset contain data that might be considered confidential (e.g., data that is protected by legal privilege or by doctorpatient confidentiality, data that includes the content of individuals non-public communications)? If so, please provide a description.* The dataset does not contain any confidential data.

*Does the dataset contain data that, if viewed directly, might be offensive, insulting, threatening, or might otherwise cause anxiety? If so, please describe why* None of the data might be offensive, insulting, threatening, or otherwise cause anxiety.

*Does the dataset relate to people? If not, you may skip the remaining questions in this section.* The dataset does not relate to people.

*Any other comments?* None.

## C   Related Work Continued

In this section, we give additional details on related work, continued from Section 2.

Multiple recent works have investigated the performance of ZC proxies in ranking architectures over different NAS benchmarks. [25] provides rank correlations and pairwise correlations of 10 ZC proxies across 7 tasks, and concludes that the relative performance of different ZC proxies highly depends on the search space. They further analyze how ZC proxies have improper biases. [48] compares 6 ZC proxies across four tasks, and further shows how `jacov` can be used to accelerate the search in predictor-based NAS. In particular, OMNI [48] combines `jacov` with *sum of training losses* [29] in the surrogate models of BANANAS and predictor-guided evolution. However, the predictor-based NAS experiments are restricted to NAS-Bench-201 and a single ZC proxy. Similar to [48], ProxyBO [33] introduces a NAS framework based on BO which uses ZC proxies to speed up NAS. It dynamically chooses whether to use a Gaussian process, `snip`, `jacov`, or `synflow` as the surrogate model in BO. Experiments were done on five tasks. Note that although the NAS method makes use of three different ZC proxies, each are used *separately* to make predictions on the performance of architectures.

Recently, NAS-Bench-Zero was introduced [2], a new benchmark based on popular computer vision models ResNet [12] and MobileNetV2 [31], and examined different characteristics of 10 ZC proxies across these search space as well as three existing search spaces. The study shows in particular that individual ZC proxies do not transfer across NAS benchmarks. They also show that voting among `synflow`, `zen`, `snip` and `synflow` is the optimal voting ZC proxy strategy. A recent overview of ZC proxies [45] computes rank correlation, pairwise correlation, and performance plots for 8 ZC proxies across 12 tasks.

Only two prior works combine the information of multiple ZC proxies together in architecture predictions [1, 2] and both only use the *voting* strategy to combine three or four ZC proxies. Our work is the first to combine ZC proxies in a nontrivial way, and the first to combine 13 ZC proxies. We also conduct analysis on the largest set of ZC proxies and benchmarks to date.

### C.1  Theoretical results for ZC proxies

While ZC proxies are starting to be used more widely today [1, 13, 45, 57], still relatively little is known about them from a theoretical standpoint. However, there have been a few works that do give theoretical results. In this section, we survey the existing theoretical results for ZC proxies.

Ning et al. gave a theoretical preference analysis for `synflow`, proving that it favors larger architectures (Section B.3 in [25]). Specifically, they prove that given an architecture, introducing a new fully-connected layer into an MLP architecture causes the `synflow` value to increase. The core of their argument is to prove the following statement: "when introducing a new fully-connected layer, the expected loss gradients with respect to the existing parameters increases." The authors also claim that the intuition for this argument should extend to convolutional neural networks. Finally, we note that our empirical results from Table 3 confirm their theoretical finding.

Shu et al. [36] attempted to give a unified, general theory for multiple ZC proxies. First, the authors prove that ZC proxy values are asymptotically similar. Specifically, they show that assuming the loss function of the neural network is $\beta$-Lipschitz continuous, and $\gamma$-Lipschitz smooth, then with high-priority, then the values of `grad_norm`, `snip`, and `grasp` are all asymptotically similar up to constants (i.e., the same under big-Oh notation) to the trace norm of the NTK matrix at initialization. This result implies that the values of these ZC proxies are highly correlated.

Next, Shu et al. establish generalization bounds for DNNs in terms of the ZC proxies. Specifically, they show that the generalization error of a DNN is at most the sum of the training error of the DNN and $O\left(\kappa/\mathcal{M}\right)$, where $\mathcal{M}$ can be set to `grad_norm`, `snip`, or `grasp`, and $\kappa$ is the condition number of the NTK matrix at initialization, i.e., given the NTK matrix $\Theta_0$, $\kappa = \lambda_{\max}(\Theta_0)/\lambda_{\min}(\Theta_0)$.

As a corollary, they also bound the generalization error in terms of the ZC proxy value and other fixed constants of the neural network, without the training error term.

Other than these results, a few works have derived new ZC proxies via a theoretical analysis or inspired by existing theories of deep learning. Shu et al. [35] introduce NASI by giving a theoretical analysis that shows the trace norm of the NTK has a similar form to gradient flow. Other theory-inspired ZC proxies include TE-NAS [4], which uses the spectrum of the NTK and the number of linear regions in the input space, and NNGP-NAS [26], which approximates the Neural Network Gaussian Process using Monte-Carlo methods.

Table 6: Spearman rank correlation for 100 architectures randomly drawn from the FBNet search space on various ZC proxies.

| ZC Proxy | fisher | flops | grad_norm | grasp | jacov | params | snip | synflow |
|---|---|---|---|---|---|---|---|---|
| Spearman | 0.2574 | **0.6484** | 0.4278 | -0.262 | -0.0895 | 0.3762 | 0.5102 | 0.4954 |

As ZC proxies gain in popularity, a further theoretical analysis is an important step in understanding their robustness on different datasets, and in designing higher-performing ZC proxies.

# D    Details from Section 4

In this section, we give additional details from Section 4.

## D.1    Details from Section 4.1: generalization

We give the full extensions of the experiments from Section 4.1. In Figure 6, for each ZC proxy and each benchmark, we compute the Spearman rank correlation (see Section 4). This is the full version of Figure 2.

In Figure 7, we compute the Pearson correlation coefficient between ZC proxy scores on pairs of benchmarks. This is the full version of Figure 3.

Next, we recompute Figure 2 using different metrics: Precision@K and BestRanking@K [2, 25]. Let $M$ denote the number of architectures, and for each architecture $a_i$ from $i \in [1, M]$, denote the rankings of the ground truth and ZC proxy-estimated scores are $r_i$ and $n_i$, respectively. Given $K$, define $A_K = \{a_i \mid n_i < KM\}$. The definitions are as follows:

$$\text{Precision@}K = \frac{\#\{i \mid r_i < K \wedge n_i < K\}}{K}$$
$$\text{BestRanking@}K = \text{argmin}_{\alpha_i \in A_K} r_i / M$$

In Figure 8, we recompute Figure 2 using Precision@K, for $K = 5, 25, 100$. In Figure 9, we recompute Figure 2 using BestRanking@K, for $K = 5, 25, 100$. Overall, we see similar trends to Figure 2, but we note that Precision@K and BestRanking@K may be more useful than Spearman in terms of NAS, since the goal of NAS is to find the very best architectures.

### D.1.1    Initial results with FBNet

While `NAS-Bench-Suite-Zero` contains 28 tasks, the majority of search spaces used were designed for research. Now, in contrast, we give initial results for FBNet [50] as a search space that has been used to achieve state-of-the-art results.

The FBNet search space consists of 22 searchable layers, with 9 operation choices each (3 filters and 3 kernel sizes), for a total of $9^{22} = 10^{21}$ architectures in the search space. The block structure is inspired by MobileNetV2 [31] and ShiftNet [51].

See Table 6 for the Spearman rank correlation values of the validation accuracy of 100 randomly drawn architectures compared to ZC proxies. Even though the FBNet search space is size $10^{21}$, some of the ZC proxies perform surprisingly well, such as `snip`, `synflow`, and `flops`. The highest-performing ZC proxy is `flops`.

## D.2    Details from Section 4.2: information theory

In this section, we give details from Section 4.2. We start with more details on the conditional entropy, including why we chose this metric, how it is computed, and how to interpret the results.

- *Why do we choose conditional entropy as the metric?*
  The conditional entropy of a random variable Y given another random variable X is

$$H(Y|X) = \mathbb{E}[-\log(p(y|x))] = -\sum_{x \in \mathcal{X}, y \in \mathcal{Y}} p(x,y) \log \frac{p(x,y)}{p(x)}, \qquad (3)$$

  for two support sets $\mathcal{X}, \mathcal{Y}$. If we assume entropy to be a measure of information, in other words uncertainty within a random variable, conditional entropy essentially captures what is left of the uncertainty after conditioning. $H(Y|X)$ also has certain desirable properties: (1). $H(Y|X) = 0$ if and only if $X$ completely determines the value of $Y$; (2). $H(Y|X) = H(Y)$ if and only if $X$ and $Y$ are completely independent; and (3). $H(Y|X_1, X_2) = H(Y, X_1, X_2) - H(X_1, X_2)$. We can then easily calculate conditional entropy when conditioning on multiple random variables, and use it as a metric for uncertain information.

- *Discretization of ZC proxy scores and ground-truth accuracies.*
  Calculating conditional entropy as prescribed above requires that all random variables be discrete, which is not the case for raw validation accuracies and ZC proxy scores. Implementation wise, we discretize all the float values and use Sturge's rule [32] as a heuristic to choose the number of bins for discretization:

$$n_{\text{bins}} = \text{round}(1 + 3.322 * \log(N))), \text{where } N \text{ is the sample size.} \qquad (4)$$

  Therefore, information about $Y$ does not reveal the exact validation accuracy but rather the interval in which the value falls.

- *Interpreting the information gain heatmap.*
  The information gain heatmap shows how much the conditional entropy of $y|z_{i_1}$ decreases to $y|z_{i_1}, z_{i_2}$ as the scores of ZC proxy on each column ($z_{i_2}$) is revealed, given that we already know the scores of ZC proxy on each row ($z_{i_1}$). For instance, on Figure 4 (top right), the value 1.42 on the second row, first column shows that $H(y|scores(\text{synflow}) - H(y|scores(\text{synflow}), scores(\text{epe\_nas})) = 1.42$. Note that (1). all values on the diagonal are 0.0 because no information is gained when we add a copy of the existing ZC proxy scores; (2). The heatmap is **not** symmetric like pairwise conditional entropy. The order in which conditioning is applied affects the amount of information gain, i.e. $\mathbf{IG}(y|z_{i_1}, z_{i_2}) \neq \mathbf{IG}(y|z_{i_2}, z_{i_1})$; (3). **IG** measures how much one ZC proxy's information complements that of another for determining the ground-truth accuracy. It does **not** serve as a direct indicator of the quality of individual ZC proxy themselves.

- *Interpreting the entropy vs. number of ZC proxies plot.*
  Conditional entropy monotonically decreases as we condition the validation accuracy, $y$, on an increasing amount of ZC proxy scores, $z_{i_1}, \ldots z_{i_k}$, which always brings in additional information. In most cases, marginal **IG** drastically decreases as the amount of ZC proxies $k$ reaches 4, but this is only true if the proxies are chosen strategically, using either a greedy or a brute-force minimization approach. For the majority of benchmarks, the less computationally intensive greedy strategy matches up to the brute-force strategy. On the other hand, randomly choosing the ZC proxies does not have stable performance and could be suboptimal, such as on NAS-Bench-201 + CIFAR-100 in Figure 4 (bottom middle).

For completion, in Figure 10, we plot the average pairwise correlation for all pairs of ZC proxies.

In Figures 11, 12, 13, 14, 15, we show all the conditional entropy and information gain heatmaps, in addition to the entropy vs. number of ZC proxies plots for all benchmark, dataset pairs. Note that for TransNAS-Bench-101, there are no results for `epe_nas` because it is not defined on non-classification tasks. Similarly, `synflow` returns 0.0 for certain non-classification tasks such as the ones in TransNAS-Bench-101, so we also removed `synflow` from the TransNAS-Bench-101 plots.

While the conditional entropy and information gain plots from Figure 4 was computed using Equation 4 to compute the number of bins, we also run the same experiment using a different discretization strategy: the bin dividers are computed based on percentages of the data. See Figure 16 (top). While the scales differ, we see largely the same trends. For example, there is still a cluster among `nwot`, `flops`, `l2_norm`, `zen`, and `params`. This suggests that this analysis is robust to the two different discretization strategies. Next, we also re-run the experiment on conditional entropy vs. $k$ from Figure 4 using the top 1000 architectures only, which may be important in the context of NAS, since NAS is concerned with finding the *best* architectures. See Figure 16 (bottom). We find that the random ordering performs comparatively better, predictably implying that it is harder to distinguish architectures that are in the top 1000 vs. randomly drawn architectures.

Table 7: Pearson correlation coefficients between predictors and bias metrics (in bold) on different datasets, for the most and least biased ZC proxies on each search space and task. For example, for the **Conv:pool** bias on NB201-CF10, `synflow` is most biased, with a correlation of 0.76, while `grasp` is least biased (in terms of absolute value), with a correlation of -0.01.

| Name | Conv:pool | | Cell size | | Num. skip connections | | Num. parameters | |
|---|---|---|---|---|---|---|---|---|
| | Most biased | Least biased | Most biased | Least biased | Most biased | Least Biased | Most biased | Least biased |
| NB101-CF10 | synflow 0.76 | grasp -0.01 | n/a | n/a | n/a | n/a | nwot 0.74 | epe_nas -0.02 |
| NB201-CF10 | l2_norm 0.87 | grasp 0.01 | synflow 0.57 | grasp -0.02 | l2_norm -0.41 | grasp -0.01 | l2_norm 0.70 | grasp 0.00 |
| NB201-CF100 | l2_norm 0.87 | grasp 0.01 | synflow 0.57 | grasp -0.01 | l2_norm -0.41 | grasp -0.01 | l2_norm 0.70 | fisher 0.01 |
| NB201-IM | l2_norm 0.87 | grasp 0.01 | synflow 0.58 | grasp 0.01 | l2_norm -0.41 | grasp -0.01 | l2_norm 0.70 | grasp 0.01 |
| NB301-CF10 | params 0.78 | fisher 0.01 | n/a | n/a | flops -0.35 | epe_nas 0.01 | zen 0.99 | epe_nas -0.01 |
| TNB101_MICRO-JIGSAW | n/a | n/a | l2_norm 0.70 | grasp -0.02 | plain 0.50 | grasp -0.01 | l2_norm 0.64 | grasp 0.02 |
| TNB101_MICRO-SCENE | n/a | n/a | l2_norm 0.70 | fisher 0.07 | plain 0.49 | grasp -0.10 | snip 0.64 | grasp -0.04 |
| TNB101_MICRO-OBJECT | n/a | n/a | l2_norm 0.70 | fisher -0.08 | plain 0.49 | grasp -0.06 | l2_norm 0.64 | grasp -0.02 |
| TNB101_MICRO-AUTOENC | n/a | n/a | l2_norm 0.70 | grasp -0.02 | grad_norm -0.46 | grasp -0.03 | l2_norm 0.64 | grasp 0.02 |
| TNB101_MICRO-NORMAL | n/a | n/a | l2_norm 0.70 | plain 0.01 | snip -0.45 | grasp -0.01 | l2_norm 0.64 | plain 0.00 |
| TNB101_MICRO-ROOM | n/a | n/a | l2_norm 0.70 | fisher 0.10 | plain 0.45 | jacov 0.14 | l2_norm 0.64 | grasp -0.01 |
| TNB101_MICRO-SEGMENT | n/a | n/a | l2_norm 0.70 | grasp 0.00 | grad_norm -0.43 | grasp 0.01 | l2_norm 0.64 | grasp -0.01 |
| TNB101_MACRO-JIGSAW | n/a | n/a | n/a | n/a | n/a | n/a | l2_norm 0.89 | plain 0.04 |
| TNB101_MACRO-SCENE | n/a | n/a | n/a | n/a | n/a | n/a | l2_norm 0.90 | plain 0.05 |
| TNB101_MACRO-OBJECT | n/a | n/a | n/a | n/a | n/a | n/a | l2_norm 0.89 | plain 0.05 |
| TNB101_MACRO-AUTOENC | n/a | n/a | n/a | n/a | n/a | n/a | l2_norm 0.89 | plain 0.01 |
| TNB101_MACRO-NORMAL | n/a | n/a | n/a | n/a | n/a | n/a | l2_norm 0.89 | grasp -0.02 |
| TNB101_MACRO-ROOM | n/a | n/a | n/a | n/a | n/a | n/a | l2_norm 0.89 | grasp 0.00 |
| TNB10_MACRO-SEGMENT | n/a | n/a | n/a | n/a | n/a | n/a | l2_norm 0.89 | plain 0.00 |

## D.3 Details from Section 4.3: biases

In this section, we give details from Section 4.3. In Table 7, for each bias metric we assess, we show the ZC proxies with the highest and lowest absolute correlation for each search space and dataset, if applicable. For the number of parameters bias, we do not consider the ZC proxies of `params` and `flops` since they trivially have 1.00 correlation. Note that operation biases are not available in TransNASBench101-Macro because the search space is architecture-level. This is an extension of Table 3.

## D.4 NAS-Bench-Suite-Zero Speedup Details

Here we show statistics on how our benchmark speeds up NAS experiments previously done with NAS-Bench-Suite by orders of magnitude. See Table 8.

Table 8: Runtimes (on an Intel Xeon Gold 6242 CPU) for all types of experiments done in this paper, with and without `NAS-Bench-Suite-Zero`. The runtimes of the experiments with NBSuite are computed by using the average training times for randomly drawn architectures from each search space in NBSuite.

| Experiment | With NBSuite (approx.) | With NBSuite + NBSuite-Zero | Speedup |
|---|---|---|---|
| Mutual information study | 158.2 hours | 124.1 seconds | 4592× |
| Architecture bias study | 6956 hours | 14.8 seconds | 1776003× |
| Standalone XGBoost+ZC, 100 trials | 1033 hours | 100 seconds | 37180× |
| BANANAS+ZC, 100 trials | 4694 hours | 4260 seconds | 3967× |
| NPENAS+ZC, 100 trials | 1033 hours | 3470 seconds | 1071× |

Table 9: Average Spearman rank correlations between XGBoost predictions and validation accuracies, for each benchmark, across three different experiments: *Encoding* uses only the encoding of the model, *ZC* uses only the ZC features, and *Both* concatenates ZC features to the encoding of the model. 100 models were used to train XGBoost.

| Features
Benchmark | Encoding | ZC | Both | % Improvement (ZC) | % Improvement (Both) |
|---|---|---|---|---|---|
| NB101-CF10 | 0.546 | 0.708 | 0.718 | 29.67 | 31.50 |
| NB201-CF10 | 0.622 | 0.905 | 0.906 | 45.50 | 45.66 |
| NB201-CF100 | 0.640 | 0.907 | 0.908 | 41.71 | 41.87 |
| NB201-IMGNT | 0.683 | 0.879 | 0.883 | 28.70 | 29.28 |
| NB301-CF10 | 0.314 | 0.405 | 0.465 | 28.98 | 48.09 |
| TNB101_MACRO-AUTOENC | 0.673 | 0.831 | 0.837 | 23.48 | 24.37 |
| TNB101_MACRO-JIGSAW | 0.809 | 0.706 | 0.809 | -12.73 | 0.00 |
| TNB101_MACRO-NORMAL | 0.617 | 0.710 | 0.716 | 15.07 | 16.05 |
| TNB101_MACRO-OBJECT | 0.736 | 0.840 | 0.843 | 14.13 | 14.54 |
| TNB101_MACRO-ROOM | 0.683 | 0.589 | 0.707 | -13.76 | 3.51 |
| TNB101_MACRO-SCENE | 0.832 | 0.891 | 0.899 | 7.09 | 8.05 |
| TNB101_MACRO-SEGMENT | 0.900 | 0.807 | 0.876 | -10.33 | -2.67 |
| TNB101_MICRO-AUTOENC | 0.714 | 0.754 | 0.803 | 5.60 | 12.46 |
| TNB101_MICRO-JIGSAW | 0.585 | 0.730 | 0.743 | 24.79 | 27.01 |
| TNB101_MICRO-NORMAL | 0.657 | 0.801 | 0.809 | 21.92 | 23.14 |
| TNB101_MICRO-OBJECT | 0.637 | 0.733 | 0.752 | 15.07 | 18.05 |
| TNB101_MICRO-ROOM | 0.582 | 0.843 | 0.844 | 44.85 | 45.02 |
| TNB101_MICRO-SCENE | 0.710 | 0.849 | 0.866 | 19.58 | 21.97 |
| TNB101_MICRO-SEGMENT | 0.767 | 0.886 | 0.897 | 15.51 | 16.95 |

# E    Details from Section 5

In this section, we give the full details from Section 5.

We start by presenting the complete standalone predictor experiments. In Section 5, we mentioned that on NAS-Bench-201 CIFAR-100, the Spearman rank correlation of XGBoost predictions trained on 100 randomly sampled architectures and averaged over 100 trials, improves from 0.640 to 0.908 when 13 ZC proxies are added. Now, we present the results of this same experiment for all benchmarks. See Table 9. We see that the large improvement is consistent across the board. We also run the same experiment when XGBoost is trained on 1000 randomly sampled architectures. See Table 10. Even though the predictions with the original XGBoost already have high rank correlation, we show that ZC proxies improve the performance even more.

## E.1    Feature importances of ZC proxies

In this section, we train an XGBoost surrogate model on 100 and 1000 randomly drawn architectures using the ZC proxies as features, and then we plot feature importances for each feature. The feature importance is calculated by the the number of times a feature is used to split the data across all trees (the default feature importance method in the XGBoost library [3]). See Figures 20 and 21 for the results with a training set size of 100 and 1000, respectively.

Table 10: Average Spearman rank correlations between XGBoost predictions and validation accuracies, for each benchmark, across three different experiments: *Encoding* uses only the encoding of the model, *ZC* uses only the ZC features, and *Both* concatenates ZC features to the encoding of the model. 1000 models were used to train XGBoost.

| Features Benchmark | Encoding | ZC | Both | % Improvement (ZC) | % Improvement (Both) |
|---|---|---|---|---|---|
| NB101-CF10 | 0.748 | 0.811 | 0.851 | 8.42 | 13.77 |
| NB201-CF10 | 0.890 | 0.954 | 0.961 | 7.19 | 7.98 |
| NB201-CF100 | 0.906 | 0.953 | 0.959 | 5.19 | 5.85 |
| NB201-IMGNT | 0.922 | 0.948 | 0.957 | 2.82 | 3.80 |
| NB301-CF10 | 0.678 | 0.496 | 0.705 | -26.84 | 3.98 |
| TNB101_MACRO-AUTOENC | 0.890 | 0.903 | 0.917 | 1.46 | 3.03 |
| TNB101_MACRO-JIGSAW | 0.812 | 0.801 | 0.856 | -1.35 | 5.42 |
| TNB101_MACRO-NORMAL | 0.692 | 0.759 | 0.764 | 9.68 | 10.40 |
| TNB101_MACRO-OBJECT | 0.846 | 0.880 | 0.888 | 4.02 | 4.96 |
| TNB101_MACRO-ROOM | 0.741 | 0.731 | 0.793 | -1.35 | 7.02 |
| TNB101_MACRO-SCENE | 0.936 | 0.936 | 0.953 | 0.00 | 1.82 |
| TNB101_MACRO-SEGMENT | 0.951 | 0.920 | 0.952 | -3.26 | 0.11 |
| TNB101_MICRO-AUTOENC | 0.838 | 0.815 | 0.861 | -2.74 | 2.74 |
| TNB101_MICRO-JIGSAW | 0.768 | 0.827 | 0.833 | 7.68 | 8.46 |
| TNB101_MICRO-NORMAL | 0.816 | 0.850 | 0.864 | 4.17 | 5.88 |
| TNB101_MICRO-OBJECT | 0.806 | 0.841 | 0.858 | 4.34 | 6.45 |
| TNB101_MICRO-ROOM | 0.874 | 0.943 | 0.947 | 7.89 | 8.35 |
| TNB101_MICRO-SCENE | 0.862 | 0.929 | 0.943 | 7.77 | 9.40 |
| TNB101_MICRO-SEGMENT | 0.921 | 0.934 | 0.948 | 1.41 | 2.93 |

### E.2 Ablation study on the number of ZC proxies

Next, we give an ablation study on the number of ZC proxies as features, for an XGBoost surrogate model trained on 1000 randomly drawn architectures. The ordering of ZC proxies is computed via the greedy method from Section 4.3. See Figure 17. We find that on all tasks, the best performance is achieved with all 13 ZC proxies (in some cases, there are ties). However, after 6-8 ZC proxies, there is only a small improvement up to the full 13 ZC proxies. This is consistent with our mutual information study from Section 4.3.

### E.3 Additional NAS results

Finally, we present more NAS results, extending the NAS results from Section 5. In Figure 18, we run BANANAS in the same setting as Section 5, on 11 benchmarks. We see that ZC proxies improve performance across the board. In Figure 19, we run the same experiment with NPENAS instead of BANANAS. Note that since NPENAS requires a mutation step, we are only able to run it on complete benchmarks: NAS-Bench-201 and TransNAS-Bench-101 (in particular, not NAS-Bench-101 or NAS-Bench-301).

## F ZC Proxy Competition

`NAS-Bench-Suite-Zero` was used successfully in the Zero Cost NAS Competition at AutoML-Conf 2022. During the competition, participants developed new, better versions of ZC proxies in the `NAS-Bench-Suite-Zero` codebase. The challenge was as follows: given $N$ models, the participant's ZC proxy will be used to rank the models for a specified task. The Kendall-Tau rank correlation is used to score the metric, averaged across three benchmarks in the test phase of the competition. The tasks in the development phase of the competition were NB201 with Ninapro and SVHN, NB301 with Ninapro and SVHN, and TNB101-Micro with Ninapro, SVHN, and Spherical-CIFAR100. The tasks in the final test phase of the competition were NB101 with CIFAR10, NB201 with ImageNet16x120, NB301 with CIFAR10, TNB101-Macro with Object Classification, and TNB101-Micro with Object Classification. The winning teams used a normalized version of `synflow`, a normalized version of `fisher`, and a product of `grad_norm` and `params`. For more information, see the competition homepage. [3]

---

[3]See `https://sites.google.com/view/zero-cost-nas-competition/home`.

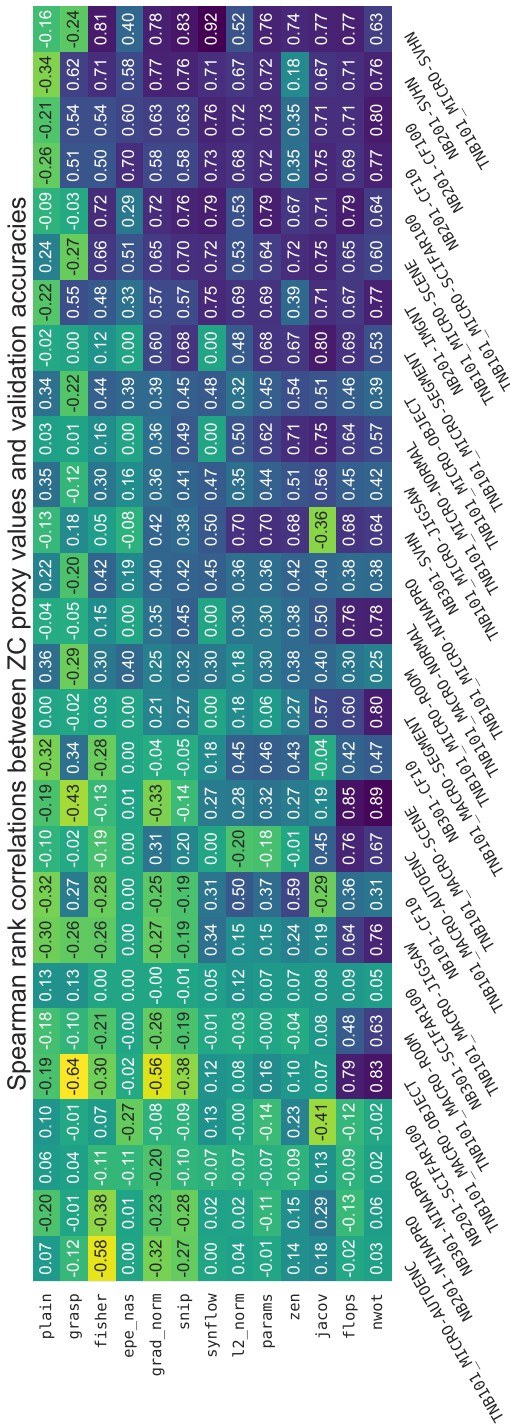

Figure 6: Spearman rank correlation coefficient between ZC proxy values and validation accuracies, for each ZC proxy and benchmark. The rows and columns are ordered based on the mean scores across columns and rows, respectively. This is the full version of Figure 2.

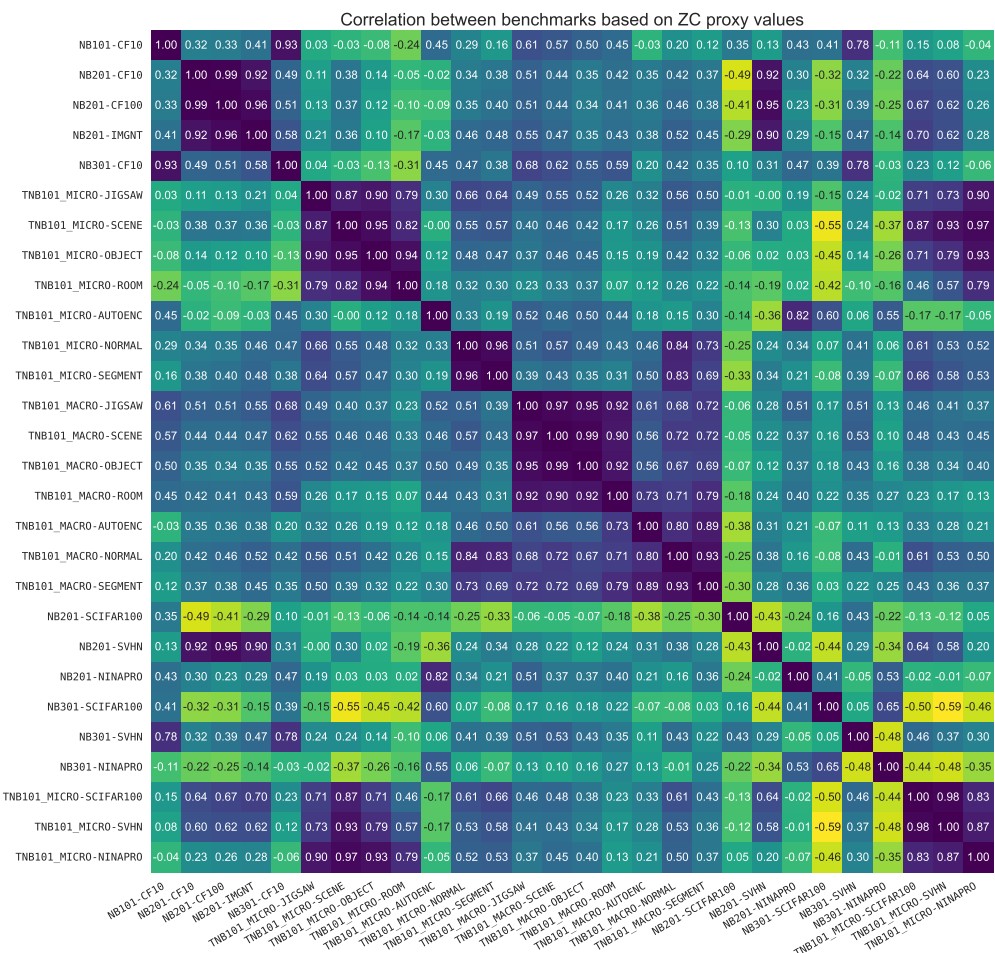

Figure 7: Pearson correlation coefficient between ZC proxy scores on pairs of benchmarks. The entries in the plot are ordered based on the mean score across each row and column. This is the full version of Figure 3.

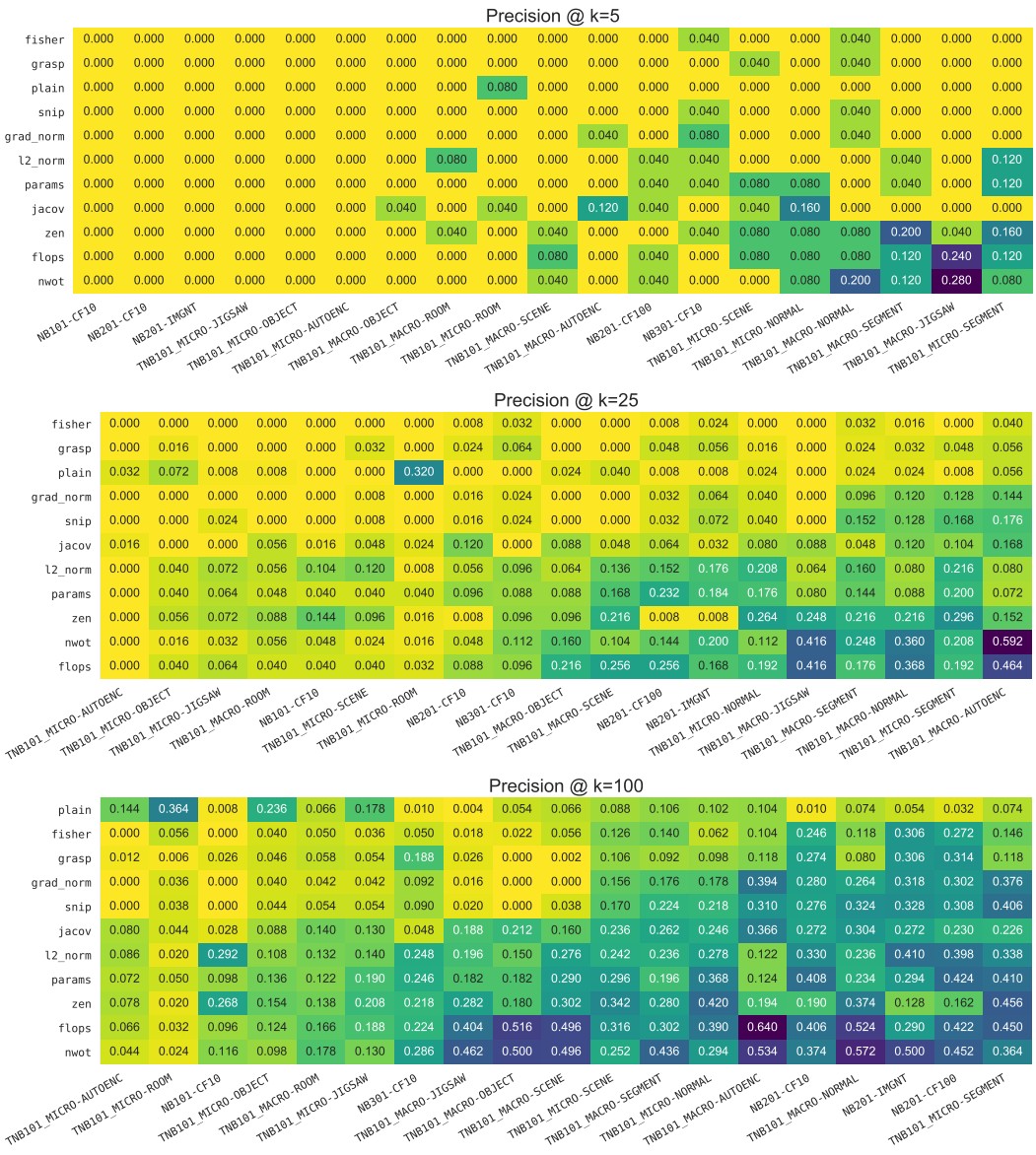

Figure 8: Precision@K between ZC proxy values and validation accuracies, for each ZC proxy and benchmark. The rows and columns are ordered based on the mean scores across columns and rows, respectively.

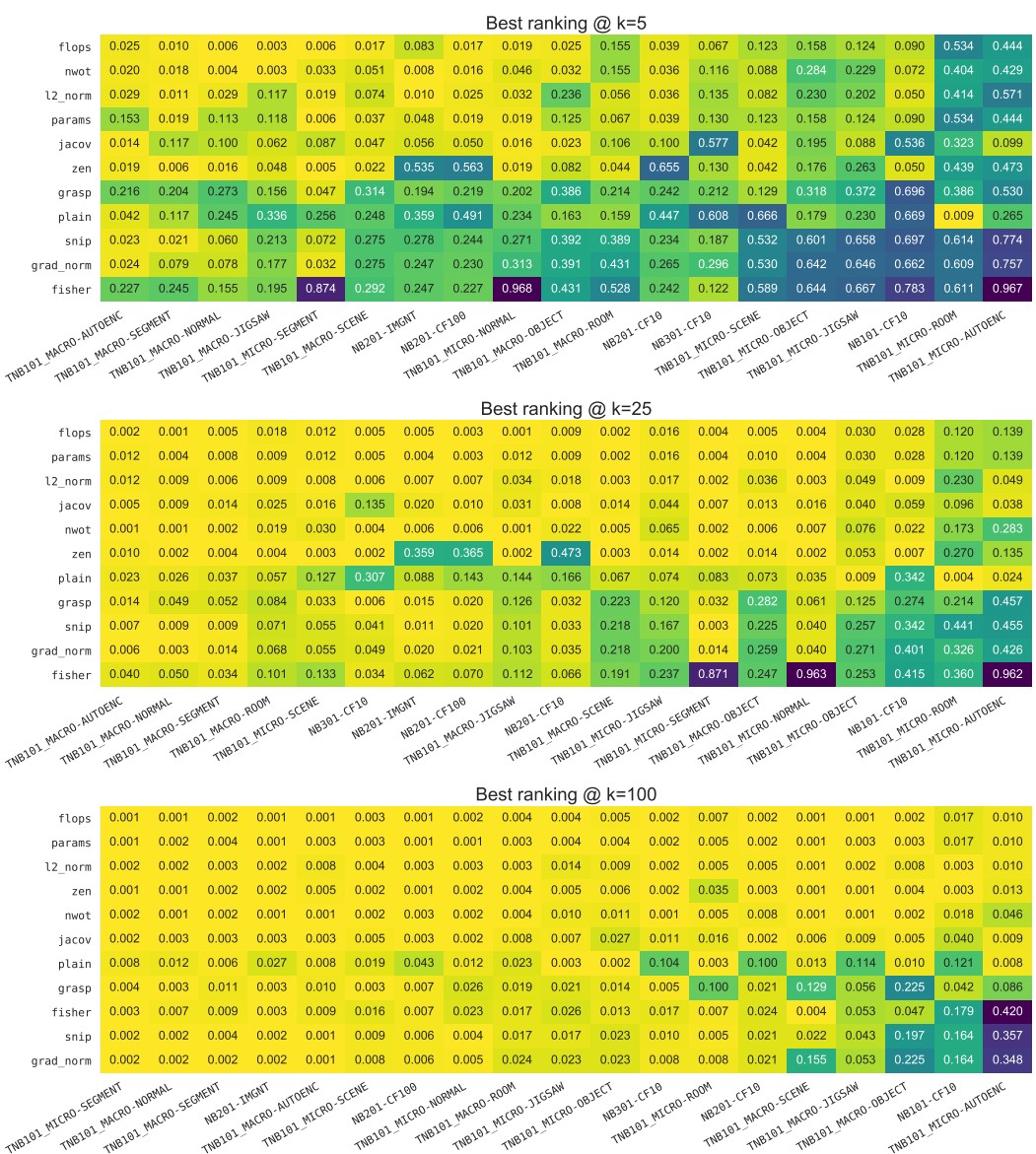

Figure 9: BestRanking@K between ZC proxy values and validation accuracies, for each ZC proxy and benchmark. The rows and columns are ordered based on the mean scores across columns and rows, respectively.

Figure 10: Pearson correlation coefficient for each pair of ZC proxies, averaged over all benchmarks. The entries in the plot are ordered based on the mean score across each row and column.

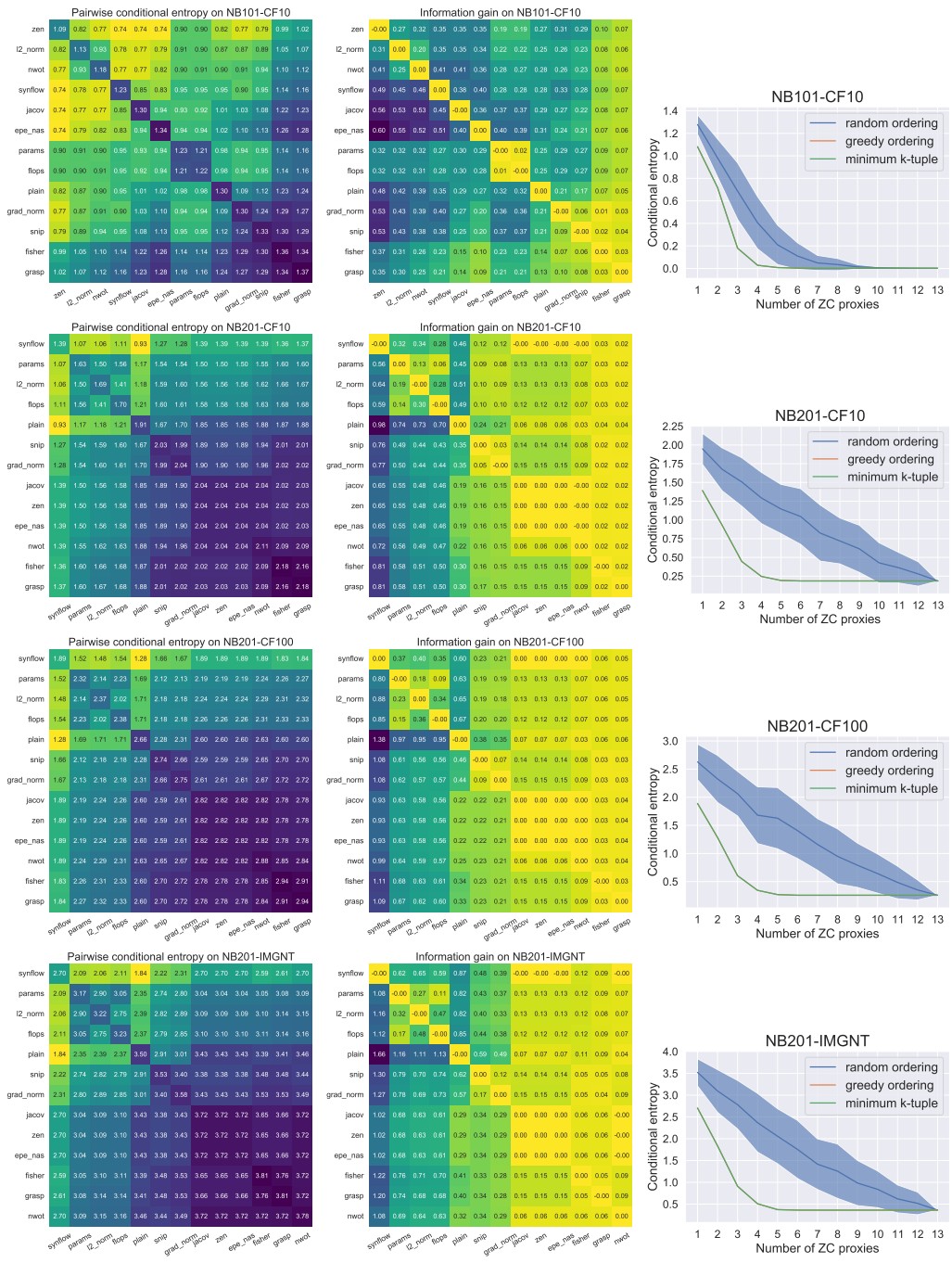

Figure 11: Conditional entropy and information gain (**IG**) for each ZC proxy pair across all search spaces and datasets (Left and Middle). Conditional entropy $H(y \mid z_{i_1}, \ldots, z_{i_k})$ vs. $k$, where the ordering $z_{i_1}, \ldots, z_{i_k}$ is selected using three different strategies (Right). (1/5)

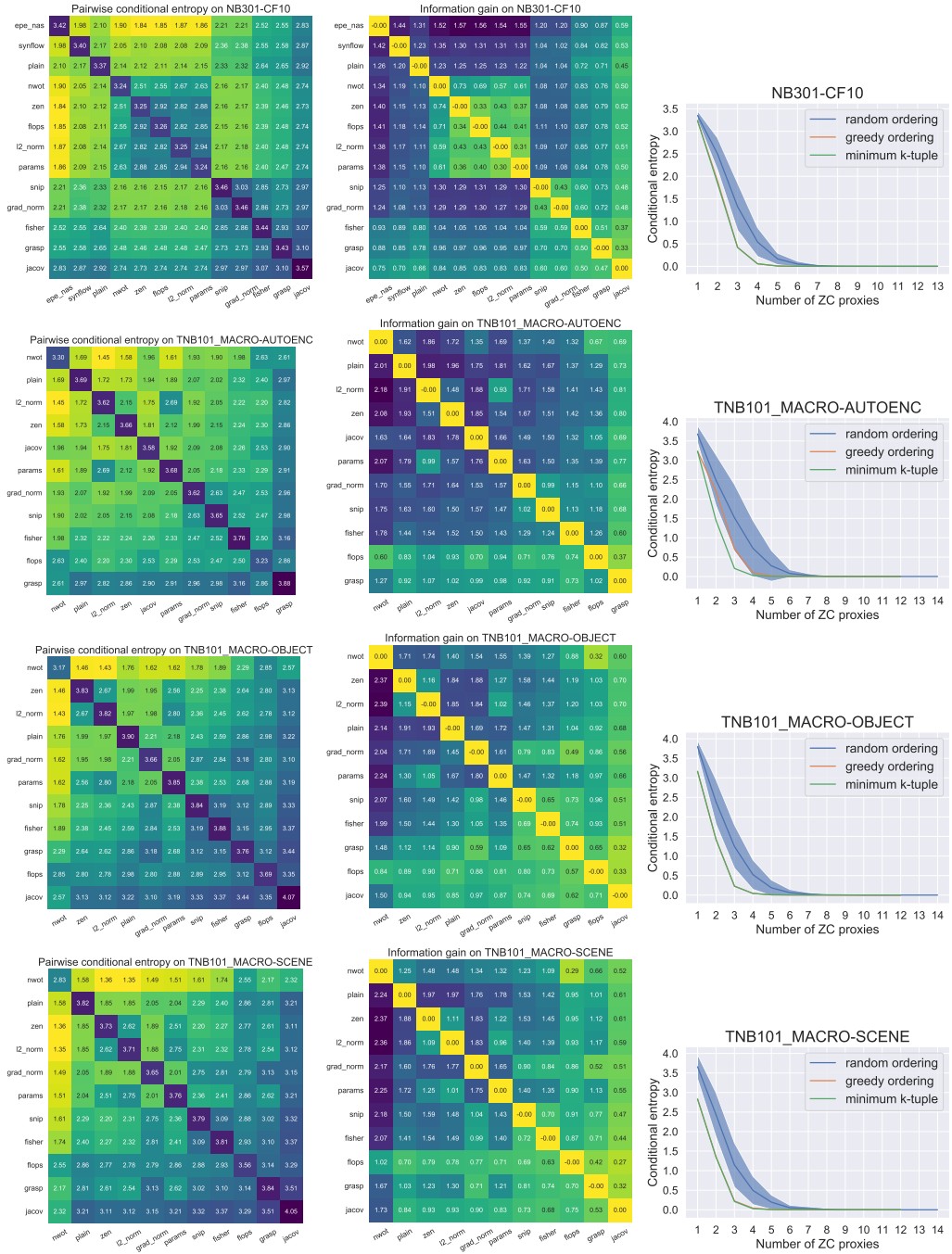

Figure 12: Conditional entropy and information gain (**IG**) for each ZC proxy pair across all search spaces and datasets (Left and Middle). Conditional entropy $H(y \mid z_{i_1}, \ldots, z_{i_k})$ vs. $k$, where the ordering $z_{i_1}, \ldots, z_{i_k}$ is selected using three different strategies (Right). (2/5)

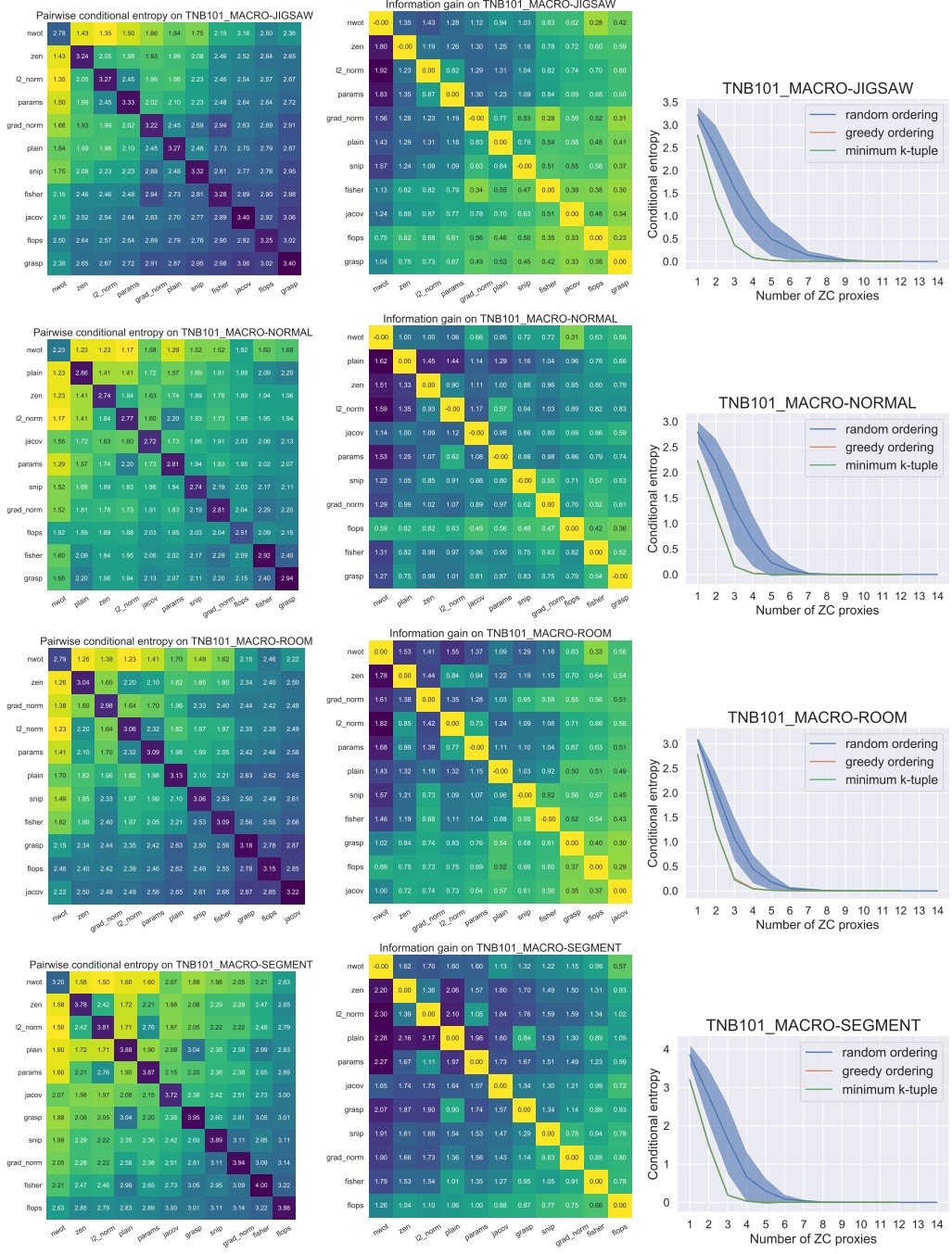

Figure 13: Conditional entropy and information gain (**IG**) for each ZC proxy pair across all search spaces and datasets (Left and Middle). Conditional entropy $H(y \mid z_{i_1}, \ldots, z_{i_k})$ vs. $k$, where the ordering $z_{i_1}, \ldots, z_{i_k}$ is selected using three different strategies (Right). (3/5)

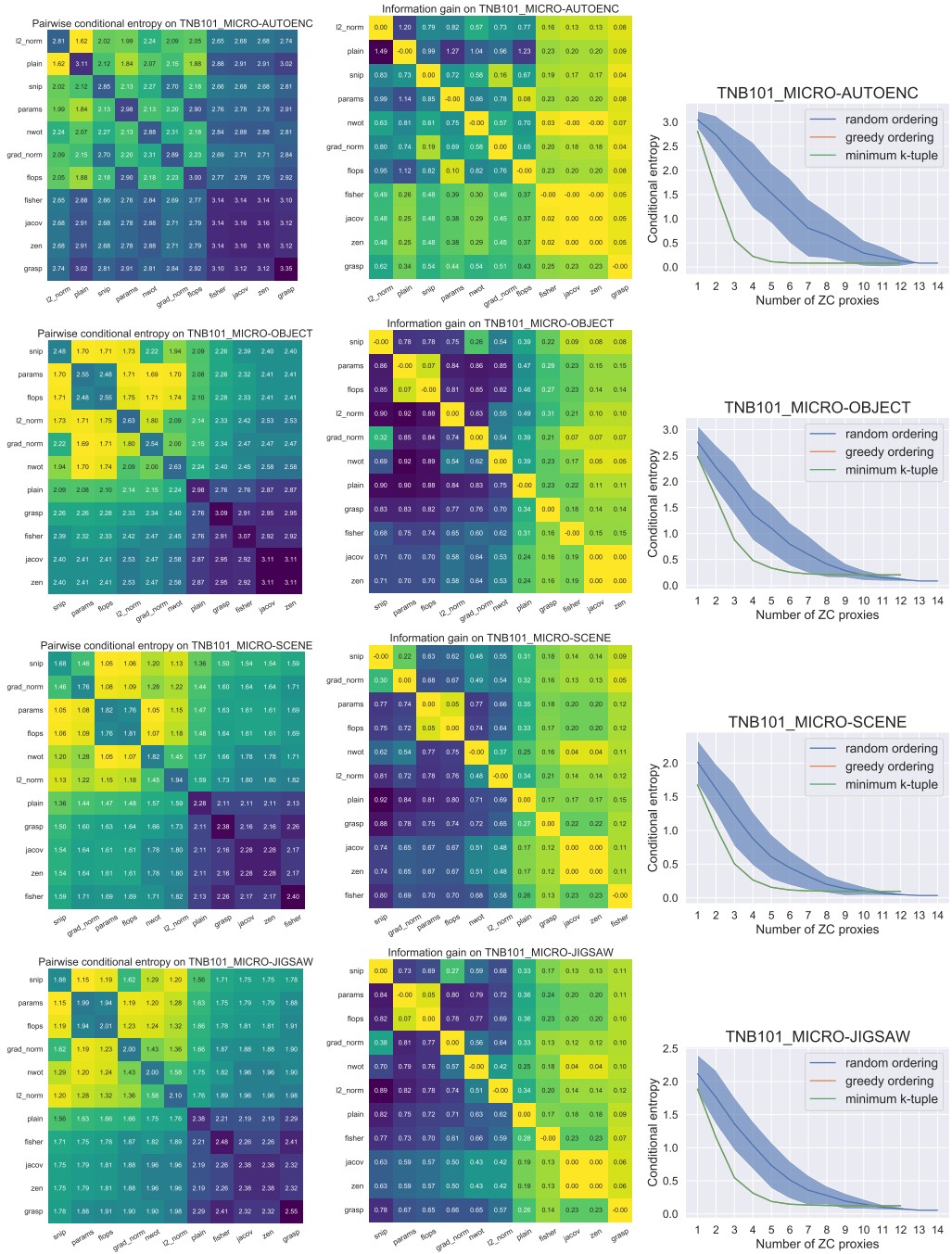

Figure 14: Conditional entropy and information gain (**IG**) for each ZC proxy pair across all search spaces and datasets (Left and Middle). Conditional entropy $H(y \mid z_{i_1}, \ldots, z_{i_k})$ vs. $k$, where the ordering $z_{i_1}, \ldots, z_{i_k}$ is selected using three different strategies (Right). (4/5)

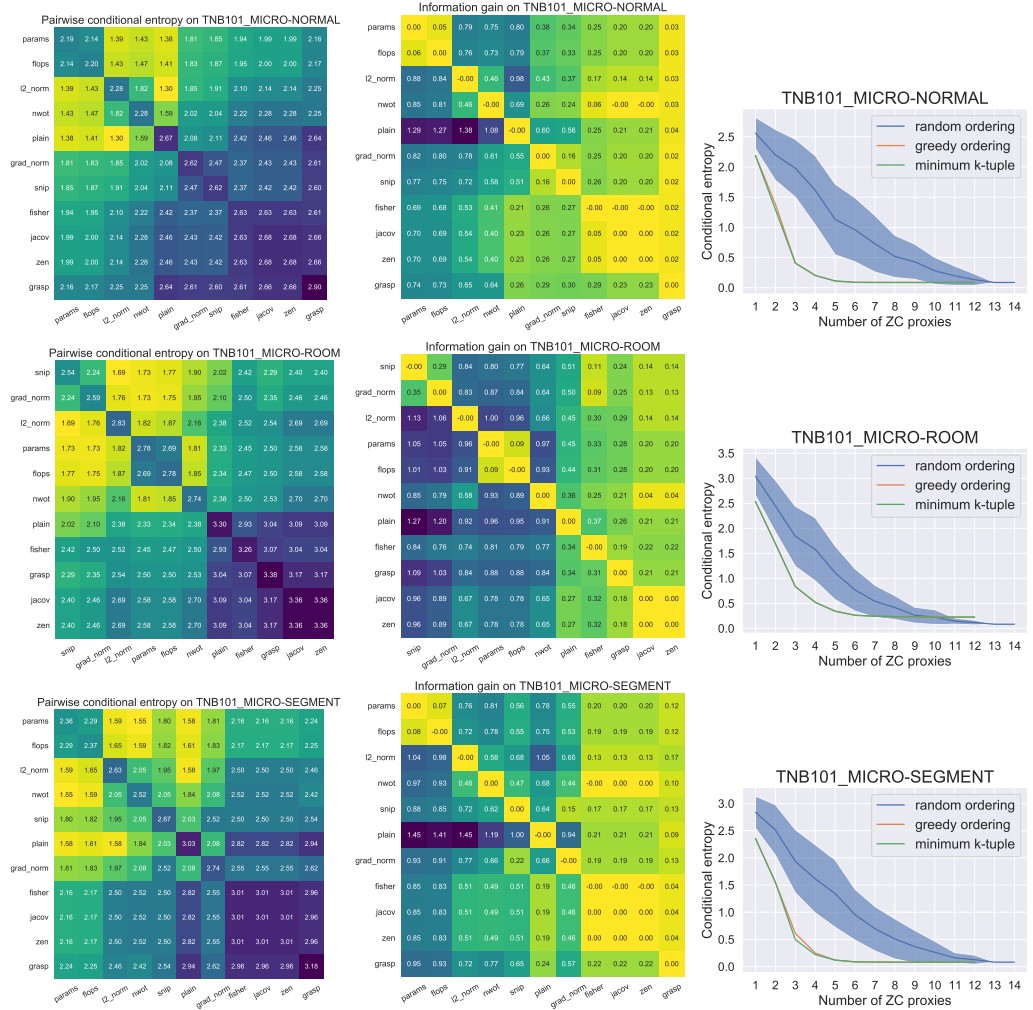

Figure 15: Conditional entropy and information gain (**IG**) for each ZC proxy pair across all search spaces and datasets (Left and Middle). Conditional entropy $H(y \mid z_{i_1}, \ldots, z_{i_k})$ vs. $k$, where the ordering $z_{i_1}, \ldots, z_{i_k}$ is selected using three different strategies (Right). (5/5)

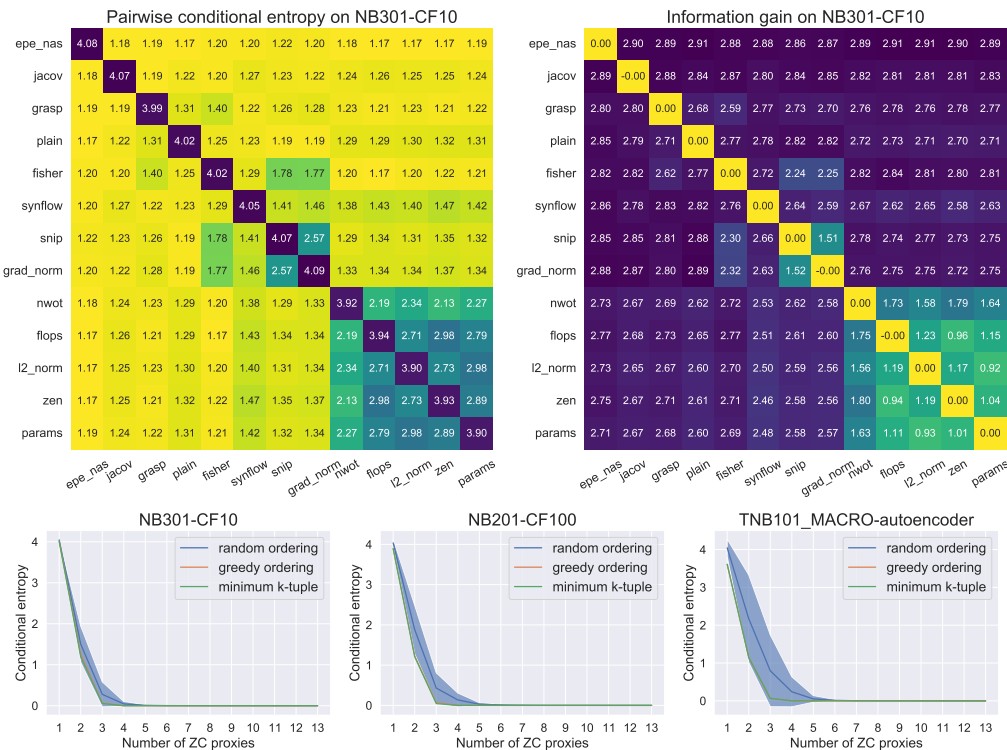

Figure 16: Given a ZC proxy pair $(i, j)$, we compute the conditional entropy $H(y \mid z_i, z_j)$ (top left), and information gain $H(y \mid z_i) - H(y \mid z_i, z_j)$ (top right). Conditional entropy $H(y \mid z_{i_1}, \ldots, z_{i_k})$ vs. $k$, where the ordering $z_{i_1}, \ldots, z_{i_k}$ is selected using three different strategies. The minimum $k$-tuple and greedy ordering significantly overlap in the first two figures (bottom). Similar to Figure 4, but using a different bin discretization strategy.

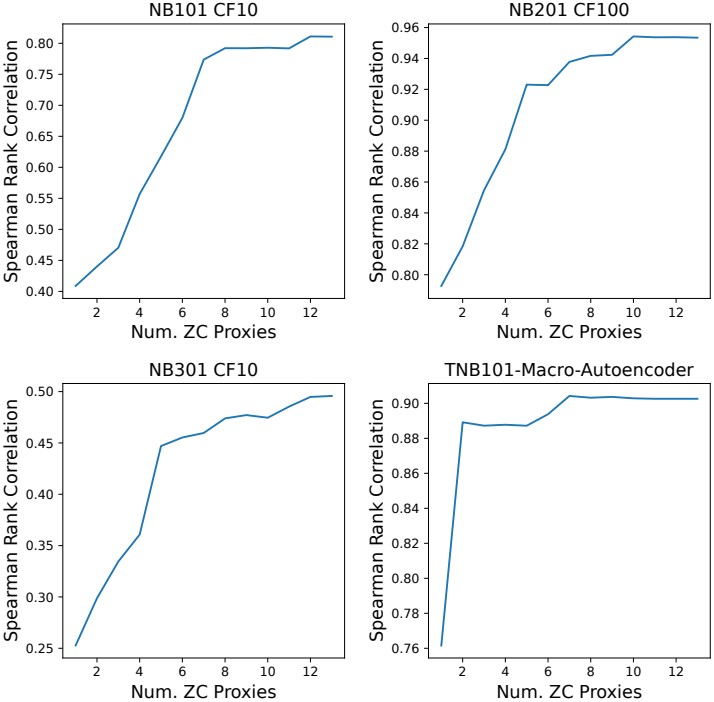

Figure 17: Ablation study on the number of ZC proxies as features vs. rank correlation performance, for an XGBoost surrogate model trained on 1000 randomly drawn architectures. The ordering of ZC proxies is computed via the greedy method from Section 4.3.

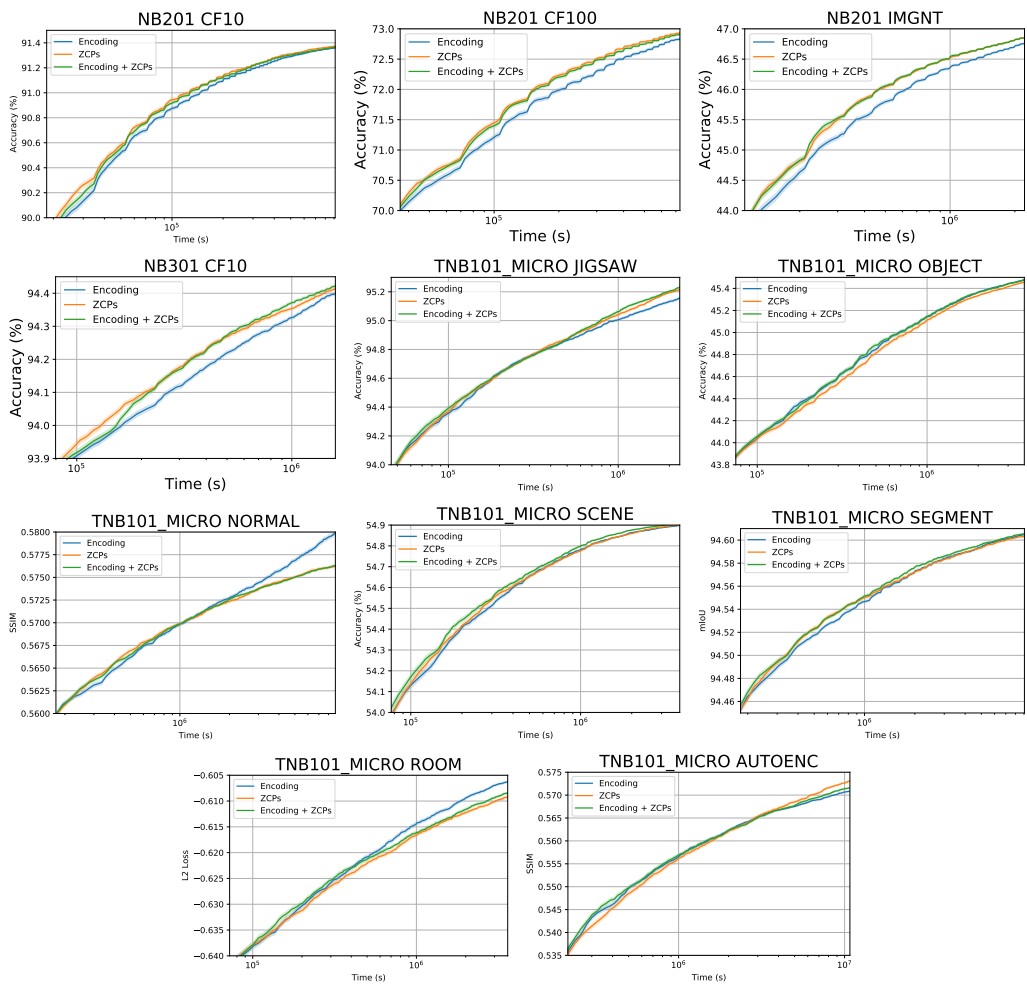

Figure 18: Performance of BANANAS with the vanilla XGBoost surrogate model vs. XGBoost using the additional ZC proxy scores (concatenated to the architecture encoding) as input.

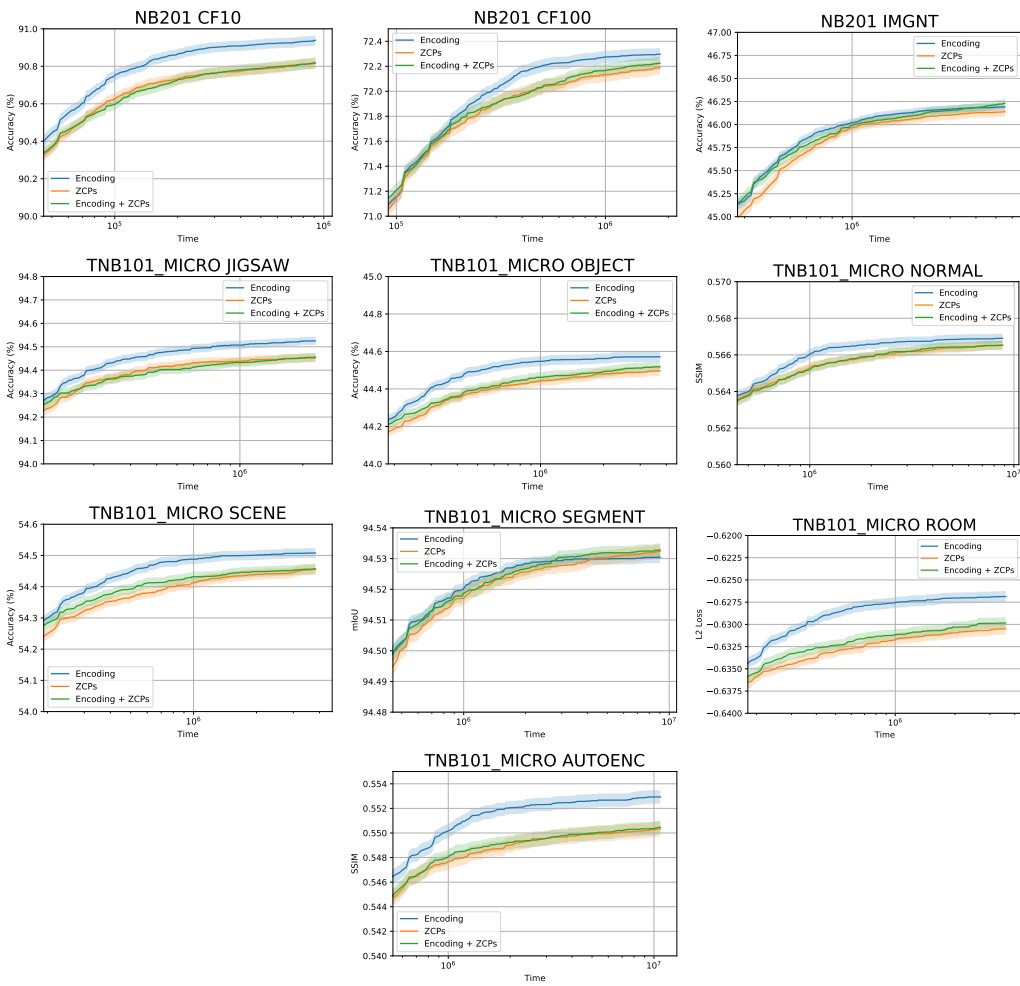

Figure 19: Performance of NPENAS with the vanilla XGBoost surrogate model vs. XGBoost using the additional ZC proxy scores (concatenated to the architecture encoding) as input.

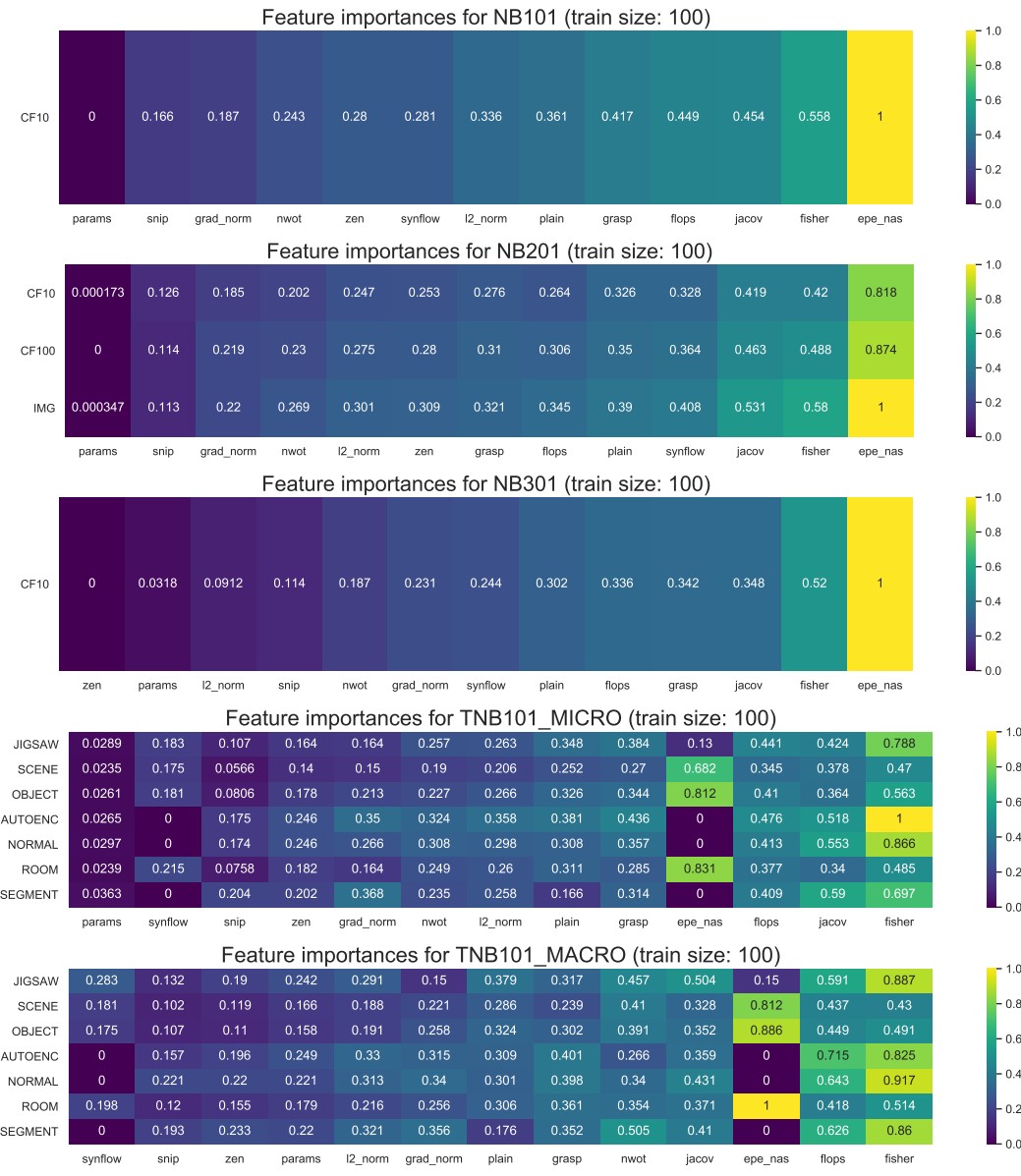

Figure 20: Feature importance values for XGBoost trained on a set of 100 architectures using ZC proxies as features.

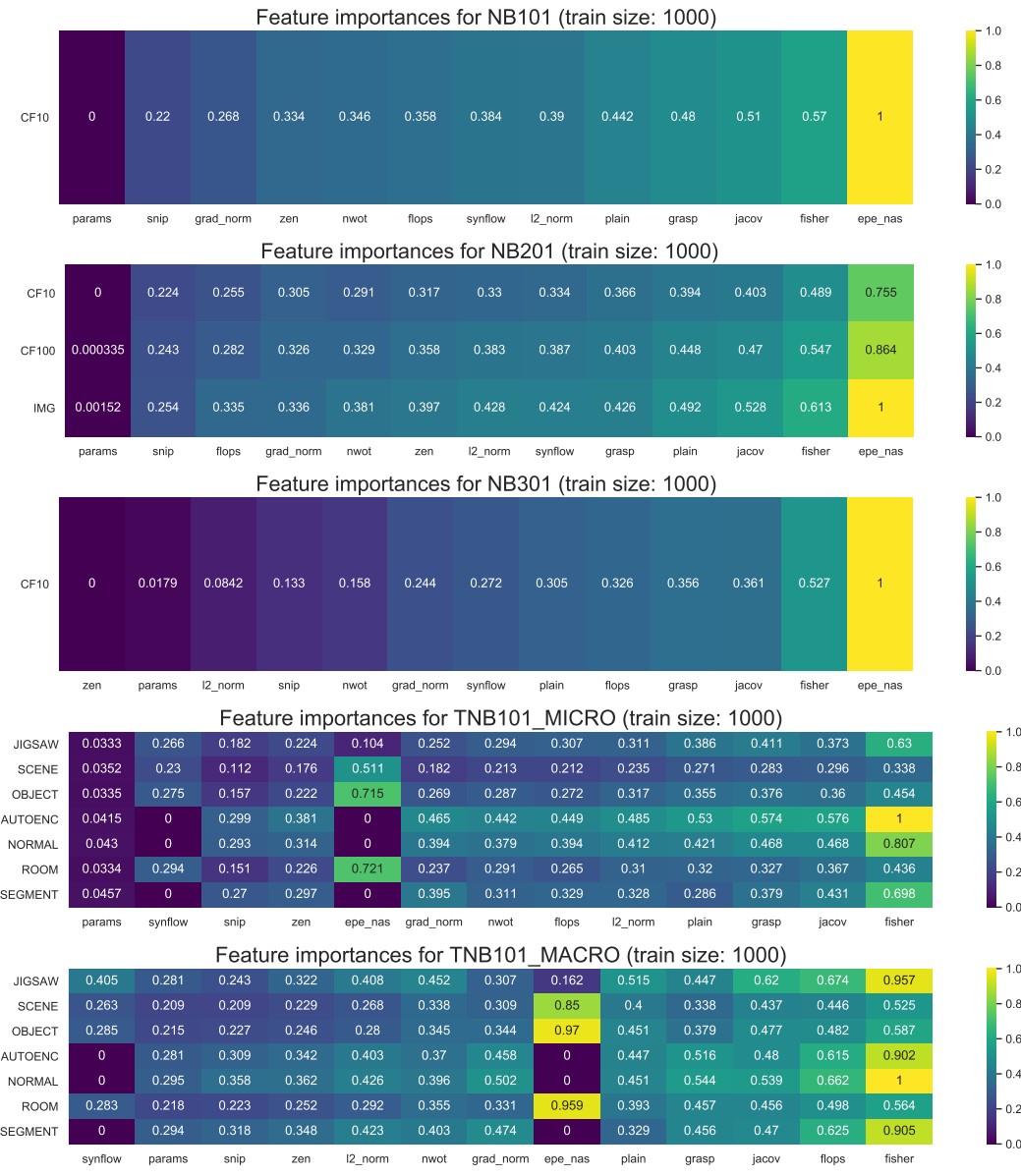

Figure 21: Feature importance values for XGBoost trained on a set of 1000 architectures using ZC proxies as features.