# OpenReview forum: "NAS-Bench-Suite-Zero: Accelerating Research on Zero Cost Proxies"
_NeurIPS.cc/2022/Track/Datasets_and_Benchmarks — NeurIPS 2022 Datasets and Benchmarks _

### Official Review · Reviewer_EsNg · 2022-07-09
**A Good NAS Benchmark on Zero-cost Proxies.**

**Rating:** 6
**Confidence:** 3

**Strengths:**

[+] The ZC proxy dataset can significantly speed up ZC proxy-based NAS experiments.

[+] The codes are accessible in github. Details of reproducibility are included.

[+] Extensive analysis of 13 ZC proxies across 28 different combinations of search spaces and tasks by studying the generalizability, bias, and mutual information among ZC proxies.

[+] The paper gives several valuable findings, e.g. combining several ZC proxies can improve the performance of NAS surrogate models and NAS algorithms.

**Weaknesses:**

[-] Although there are docs about how to reproduce the results in the paper, the interfaces using the datasets are missing. For further research, the authors are expected to provide detailed documentation and quickstart about how to use the codebase.

[-] Conducting more theoretical analysis would be better.


**Additional Feedback:**

None

**Clarity:**

Yes, the paper is well-written and easy to follow.

（Minor typos）
1. Figure 4 caption, "top right/left"
2. Line 292

**Correctness:**

Yes, all the claims are verified.


**Documentation:**

Yes, there is sufficient detail on data collection and organization, availability and maintenance, and ethical and responsible use.

**Ethics:**

No.

**Relation To Prior Work:**

Yes. Related works are discussed in Main paper Sec. 2 and Appendix Sec. C.

**Summary And Contributions:**

This paper creates a NAS benchmark NAS-Bench-Suite-Zero including 13 zero-cost(ZC) proxies, 28 tasks and 1.5 million ZC proxy scores in total. The dataset can be used to speed up ZC proxy-based NAS experiments and provide convenience for ZC proxy-based NAS research. The authors also conduct generalizability, bias and information-theoretic analysis of ZC proxies, showing that combining several ZC proxies can improve the performance of NAS surrogate models and NAS algorithms.

---

> ### Author Response · Authors · 2022-08-14
> **Thank you for the insightful feedback**
>
> We thank the reviewer for their review. We reply to each point below.
>
> **"Provide detailed documentation and quickstart about how to use the codebase"**
>
> Although we did include setup and experiment instructions [here](https://github.com/automl/NASLib/tree/zerocost#setup) and [here](https://github.com/automl/NASLib/tree/zerocost#experiments), we agree that it is crucial to easily explain our data interface, so we added a quickstart notebook [here](https://github.com/automl/NASLib/blob/zerocost/plotting/PlotCorrelations.ipynb), which users can run to easily use NAS-Bench-Suite-Zero for experiments and analysis. Thank you for this suggestion!
>
> **"Conducting more theoretical analysis would be better"**
>
> We agree that theoretical analysis would be very beneficial. We respectfully remind the reviewer that this is a track that focuses on datasets and benchmarks, so theory may be less of an emphasis. Nevertheless, we have now added a section to our paper (Appendix C.1) that surveys the theoretical analysis for ZC proxies, which we think will be very helpful for readers. We thank the reviewer for bringing this to our attention.
>
> We thank you once again for your review. We are happy to answer any follow-up questions or new comments.

---

### Official Review · Reviewer_A4Zd · 2022-07-14
**Solid work but lacking some highlights**

**Rating:** 7
**Confidence:** 5
**Correctness:** Yes
**Clarity:** Yes

**Strengths:**

1. This work is very well organized and written.
2. The authors have done more experiments and have done a good job of open source work.

**Weaknesses:**

1. This work seems to me to be just a review result, and it is more appropriate to appear in a new NAS method rather than a new benchmark, since almost all methods and datasets exist, and much of the data has already appeared in previous work.
2. Some of the data from this work is questionable, such as why it is so bad on Trans NAS.

**Additional Feedback:**

No

**Documentation:**

Yes

**Relation To Prior Work:**

Yes

**Summary And Contributions:**

This work replicates some of the typical Zero NAS on multiple NAS benchmarks.This work gives Zero NAS fair comparison data and gives some new research challenges such as on trans datasets.However, I still think the contribution of this work is not enough for the top conferences.

---

> ### Author Response · Authors · 2022-08-14
> **Thank you for the feedback**
>
> We thank the reviewer for their review. We reply to the points below.
>
> **"Just a review result"**
>
> We respectfully remind the reviewer that our contributions are the following:
> 1. We release a dataset consisting of _1.5 million_ ZC proxy evaluations (see Table 2). Until this paper, there was no existing dataset of pre-evaluated ZC proxies. This speeds up many types of NAS experiments by a factor of $10^4$ to $10^7$ (see Table 6 in the appendix for the exact numbers). And we respectfully remind the reviewer that the datasets and benchmarks track is the correct venue for this type of contribution.
> 2. We demonstrate the usefulness of our new dataset by conducting a large-scale study of the generalizability, bias, and mutual information of ZC proxies, including their performance in NAS algorithms.
>
> **"Some of the data from this work is questionable, such as why it is so bad on Trans NAS"**
>
> All of our data and code used to generate that data are open-source, which you can view and download [here](https://github.com/automl/NASLib/tree/zerocost#data). We have no stakes in how well the methods perform on TransNAS-Bench, and we believe that making *all* results public (both results that perform well, and results that perform poorly) is crucial to the advancement of science. If the reviewer has any more questions about our results, please let us know.
>
> We thank you once again for your review. We are also happy to answer any follow-up questions or new comments. Thank you!

---

> > ### Comment · Reviewer_A4Zd · 2022-08-15
> > **This benchmark is very strange and simply provides some test results on an existing benchmark.**
> >
> > The author's reply still doesn't dispel my doubts. This work only provides some evaluation results on the existing benchmarks. And the methods are also open source. If it can be considered a contribution, can those OpenMM, NNI, pytorch-imagenet and other open source projects be accepted as benchmarks by top conferences?

---

> > > ### Author Response · Authors · 2022-08-22
> > > **Thank you for clarifying**
> > >
> > > We thank the reviewer very much for further clarifying their concerns. We give a reply below.
> > >
> > > **”Can those OpenMM, NNI, pytorch-imagenet and other open source projects be accepted as benchmarks by top conferences?”**
> > >
> > > We would respectfully like to remind the reviewer of the following publications:
> > > - [OpenML Benchmarking Suites](https://openreview.net/forum?id=OCrD8ycKjG) was accepted to this same track, the NeurIPS Datasets and Benchmarks track, last year.
> > > - [An HPO benchmark based on OpenML](https://openreview.net/forum?id=O24OhmqpJtP) was also accepted to the same track last year.
> > > - [PyTorch](https://papers.nips.cc/paper/2019/file/bdbca288fee7f92f2bfa9f7012727740-Paper.pdf) was accepted to NeurIPS 2019.
> > > - [ImageNet](https://ieeexplore.ieee.org/document/5206848) was accepted to CVPR 2009.
> > >
> > > The contribution of our work is to organize and implement the various ZC proxies in the same framework, without confounding factors, create and release a dataset of size 1.5 million concerning their performances on 28 different queryable NAS benchmarks, and show its usefulness by analyzing their generalizability, bias, mutual information, and NAS performance. Such contributions, which directly help the community do better research, are absolutely in scope for this track (for example, see [HPOBench](https://openreview.net/forum?id=1k4rJYEwda-), [HPO-B](https://openreview.net/forum?id=O24OhmqpJtP), and [OpenML](https://openreview.net/forum?id=O24OhmqpJtP)).
> > >
> > > Thank you once again for responding and giving us a chance to clarify. If you have any follow-up comments, we would be very happy to continue the conversation.

---

> > > > ### Comment · Reviewer_A4Zd · 2022-08-25
> > > > **Thanks for reply**
> > > >
> > > > Thank the author for his answers. In fact, I also participated in the zero NAS competition. However, I implemented many methods in the competition, but they did not work on the evaluation server (memory overflow). I look forward to the second zero NAS competition. I have updated the score. Thank you.

---

### Official Review · Reviewer_buDu · 2022-07-24
**review of "NAS-Bench-Suite-Zero"**

**Rating:** 7
**Confidence:** 4
**Correctness:** The dataset is constructed in a sound…
**Clarity:** 1. Some citations have already been o…

**Strengths:**

1. The authors create the `NAS-Bench-Suite-Zero`, an extensible collection of 13 ZC proxies on 28 NAS benchmark tasks. Such proxies are precomputed and accessible through a unified interface.
2. The authors run a large-scale analysis of the above dataset to study the generalizability, bias, and mutual information among ZC proxies.

**Weaknesses:**

1. About the search space selection. As [3, 45] and other works including this one observe, the performance of ZC proxies highly depends on the search space and task. However, the search space used in this benchmark is far from the modern search space, such as ResNet and MobileNet [3]. This limits the significance of this work and makes observations in it less guiding to the real-world NAS applications.

2. The authors use Spearman ranking correlation to evaluate the effectiveness of ZC proxies. However, ranking correlations cannot naturally reflect the true ability in the scenarios of (constrained) NAS [3,25]. That's why other metrics such as the Precision@K, BestRanking@K [25].

3. The authors use conditional-entropy based information gain to prove the improvement of predictive power of a tuple of ZC proxies. The idea is interesting, but the IG is considering the whole distribution, which also deviates from the goal of NAS to select the top-`k` architectures.

**Additional Feedback:**

a few additional questions

1. the necessity of introducing IG in Sec. 4.2 other than using xgboost with different combination of ZC proxies (like Section 5)
2. The discretizing (D.2) to group the ground truth accuracy into `round(1 + 3.322 ∗ log(N )))` bins. The paper doesn't give detailed bin size (how many samples are in each bin), but it could be estimated (through Table 2) that the number of bins is not so large.  Could this rough estimation be accurate enough for NAS? For example, even the top bin may have more samples than the final selected. Thus, a smaller conditional entropy may not help to predict whether a sample is in the top bin.

**Documentation:**

yes

**Ethics:**

no ethics concerns.

**Relation To Prior Work:**

yes

**Summary And Contributions:**

This paper targets the ZC-NAS, which is an interesting and promising subfield of NAS, and aims to answer the important questions about the performance, generalization, and bias of ZC proxies and their combinations. To achieve this, the authors create the `NAS-Bench-Suite-Zero` codebase, precompute ZC scores on the architectures in existing benchmarks, and run a comprehensive analysis based on them.

---

> ### Author Response · Authors · 2022-08-14
> **Thank you for the constructive feedback**
>
> We thank the reviewer for their excellent suggestions. We reply to each point below.
>
> **"modern search space, such as ResNet and MobileNet"**
>
> Thank you for this suggestion. We now have initial results for FBNet [1] as the 29th task in our benchmark suite. The FBNet search space consists of 22 searchable layers, with 9 candidate blocks including MobileNetV2 [2] and ShiftNet [3] inspired blocks. Please see Appendix D.1.1 of our updated paper for more details.
>
> [1] [FBNet](https://arxiv.org/abs/1812.03443)
> [2] [MobileNetV2](https://arxiv.org/abs/1801.04381)
> [3] [ShiftNet](https://arxiv.org/abs/1711.08141)
>
> **"The authors use Spearman"**
>
> We agree, and so we have added Precision@K and BestRanking@K experiments in Appendix D.1. In fact, we would like to continue re-running experiments from the paper with the new metrics and move the Spearman results to the appendix. We agree with the reviewer that these are very informative metrics, targeted towards NAS performance.
>
> **"IG deviates from the goal of nas, to select the top-k architectures"**
>
> Thank you once again for this comment. Based on your comment, we considered a few approaches for an IG analysis that weights the top-k architectures more highly, and we decided to run the IG analysis on only the top 6% of architectures. Please see Appendix D.2 for more details.
>
> **"Some references are still pre-print"**
>
> Thank you, we have now updated all of our references to have the most up-to-date information.
>
> **"necessity of introducing IG"**
>
> An information-theoretic analysis of ZC proxies is necessary to understand the information gain and pairwise conditional entropy of ZC proxies. For example, not even a feature importance analysis of XGBoost can reliably accomplish this. However, we completely agree with your comment that IG analysis does not place a high enough emphasis on the top-k architectures, which are the most important for NAS. This is why we re-run the experiments using only the top architectures (Appendix D.2).
>
> **"Is the discretization accurate enough for NAS?"**
>
> This is a great question. Given your concern, we conduct an ablation study in which we test two different bin discretization strategies: equal-spaced bins across the min and max values, and percentages-based bin discretization (according to Sturge’s Rule).
>
> We thank you once again for raising these important points and giving constructive suggestions. We especially believe that the new addition of the FBNet search space, new analysis on top-k IG, and new experiments for Precision@K and BestRanking@K improve our work and make our work more relevant to the community. We kindly ask that you consider updating your score if you think the additions we made improve our work. We are also happy to answer any follow-up questions or new comments. Thank you!

---

> > ### Comment · Reviewer_buDu · 2022-08-27
> > **good updates**
> >
> > Thanks for the clarifying and the solid updates. I have no further questions and will update the score.

---

### Official Review · Reviewer_bXs5 · 2022-07-27
**Very well written and a great example of reproducible research**

**Rating:** 7
**Confidence:** 3

**Strengths:**

* Very detailed documentation of creation of the benchmark suite, research questions as well as experimental setup. Following best practices throughout (clearly stating infrastructure detals, hyperparameter choices, filled-out NAS best practice checklist, etc)
* Benchmark dataset access is straightforward and well documented, all code and supporting materials (e.g., figures) are included in the git repository
* Appendix contains a very detailed data sheet addressing many common questions users may have and extending even to ethical and social implications - this is outstanding
* Writing and elaboration of research questions and corresponding answers is very clear and accessible
* research findings are very thought-provoking and frequently point out clear directions for future research,  specifically:
    * Using ZC proxies as task features for meta-learning
    * Degree to which ZC information is complementary is task-dependent (motivating experimentation in this work to combine them by adding corresponding features to the XGBoost inputs)

**Weaknesses:**

* A potentially missed opportunity may consist in performing an analysis of feature importance of the ZC proxy features on the trained surrogate model. From the information-theoretic analysis we can argue that combining multiple ZC proxies is beneficial. However, it is not obvious to me from these results that we can exclude a merely marginal contribution to the speedup by individual ZC proxies. Feature importance values may indicate which (combinations of) proxies actually contribute to observed speedups.
* RQ3 answers seem a bit inconclusive, i.e. not immediately actionable. It may be helpful to outline how removing these biases would work, in a concrete case.

**Additional Feedback:**

Thank you for this insightful and inspiring work.

I have two questions (these could also just be relevant as future directions):

* The work suggests that ZC proxies may be used to assess similarities across benchmarks and serve as task features for meta-learning (see "Answer to RQ 1"). Could one turn this around and instead use a separate set of task features to learn which ZC proxies are likely to provide benefits for a given benchmark?
* What is an optimal number of ZC proxies to ensemble (this ties into my point around feature importance analysis above)? Are all 13 proxies needed? I believe this question is likely to grow in importance if the benchmark grows and additional ZC proxies get developed and integrated.

**Clarity:**

The paper is written in a very accessible way and all research questions and associated findings are very thoroughly documented (including further details in the appendix).

**Correctness:**

Construction of the dataset and decisions to include the selected 13 ZC proxies and 28 benchmarks are valid and explained well. Evaluation methods and experiment design for the benchmark experiments are appropriate and conclusions drawn from the results are valid.

**Documentation:**

A data sheet in the supplementary materials addresses all questions related to data collection, organization, availability and maintenance in great detail.
Accessibility of this data sheet could be improved by adding this information in a markdown file in the repository and referencing it directly from the top-level README file to raise awareness.
Experimental setup has been documented sufficiently and infrastructure details and hyperparameter settings have been included wherever applicable. Reproducibility of the results should therefore be guaranteed.

**Ethics:**

No ethical concerns.

**Relation To Prior Work:**

The authors do a great job situating their contributions within the prior research work in this domain.
It is clear, in what ways the new dataset differs from prior benchmarks. The new results that highlight complementary information in different ZC proxy methods as well as variability across benchmarks can confirms and explains some earlier findings referenced in this section (e.g., work cited in this one that had found a strong dependence of ZC proxy effectiveness on search space used).

**Summary And Contributions:**

Recent prior work introduced zero-cost proxies (ZC proxies) as a means to predict architecture performance to significantly speed up neural architecture search algorithms.
This work introduces a novel benchmark dataset "NAS-Bench-Suite-Zero" to study 13 ZC proxies across 28 tasks.
Main contributions and findings are:
* Introduced largest (to date) benchmark dataset for ZC proxies, streamlining and speeding up experimentation
* Information-theoretic analysis of ZC proxy behavior yielding finding that they capture complementary information
* Evaluation showing that incorporating all 13 ZC proxies into NAS surrogate models improves their predictive performance significantly

---

> ### Author Response · Authors · 2022-08-14
> **Thank you for the great feedback**
>
> We thank the reviewer for their positive feedback and excellent suggestions. We reply to the points below.
>
> **"Performing an analysis of feature importance on the trained surrogate"**
>
> We agree that this is a great idea. We have now added this experiment in Appendix E.1. We note that feature importance and information theory analysis both have their strengths and weaknesses. For example, if multiple features share the same information, XGBoost might only pick one of them in a deterministic manner, whereas information theory can compute the exact level of complementary vs. shared information for any set of ZC proxies. We agree that presenting both feature importances and information theory gives the most complete analysis.
>
> **"Section 4.3 is not actionable"**
>
> This is a great suggestion. We have now included a concrete method for reducing bias in ZC proxies. Please see Appendix D.3.1. We find that we can (1) reduce bias to zero at the cost of performance, or (2) reduce bias so that it matches the bias of the ground truth validation accuracy, which often improves performance. There are still a few details before this can be efficient (e.g., how to quickly determine the bias of the ground truth architectures), but this can have important ramifications for future ZC proxy design. Thank you for the comment!
>
> **"Learn which ZC proxies are likely to provide benefits for a given benchmark"**
>
> This is a great question for future work. In order to make this an actionable strategy that does not overfit, it may require additional benchmarks.
>
> **"What is the optimal number of ZC proxies to ensemble"**
>
> The question is similar to our Fig. 4 that computed conditional entropy vs. num. ZC proxies, where we see a sharp taper after 4-5 ZC proxies (if using an optimal or greedy ordering). Therefore, we would conjecture that there is not much benefit after 4-5 when using a greedy ordering, but still benefit of using more if ZC proxies are chosen randomly. We are working to confirm this now.
>
> **"Accessibility of our datasheet"**
>
> Thanks for this suggestion. We have now added full documentation info to the [readme](https://github.com/automl/NASLib/tree/zerocost#documentation), including our
> [datasheet](https://github.com/automl/NASLib/blob/zerocost/docs/DATASHEET.md),
> [license](https://github.com/automl/NASLib/blob/zerocost/LICENSE),
> [author responsibility](https://github.com/automl/NASLib/blob/zerocost/docs/AUTHOR_RESPONSIBILITY.md),
> [code of conduct](https://github.com/automl/NASLib/blob/zerocost/docs/CODE_OF_CONDUCT.md), and
> [maintenance plan](https://github.com/automl/NASLib/blob/zerocost/docs/MAINTENANCE_PLAN.md).
>
> We thank the reviewer once again for these great suggestions, which we believe have further improved our work. Please let us know if you have any follow-up questions or any additional suggestions.

---

> > ### Author Response · Authors · 2022-08-22
> > **Update on the optimal number of ZC proxies**
> >
> > We give a quick update.
> >
> > **"What is the optimal number of ZC proxies to ensemble"**
> >
> > We have now finished this experiment (Appendix E.2). The best performance is achieved with all 13 ZC proxies. However, after 6-8 ZC proxies, there is only a small improvement up to the full 13 ZC proxies. This is consistent with our mutual information study from Section 4.2, and not too far off of our [conjecture](https://openreview.net/forum?id=yWhuIjIjH8k&noteId=Q9CDHVxah3Y) above.
> >
> > Thank you again for this suggestion, and please let us know if you have any questions about this new experiment or anything else from our initial reply.

---

### Official Review · Reviewer_o5Zx · 2022-07-28

**Rating:** 7
**Confidence:** 4
**Clarity:** The paper is well written and easy to…

**Strengths:**

1. The collection of benchmarks and ZC proxies that unifies and accelerates research on ZC proxies – a promising new sub-field of NAS – by enabling orders-of-magnitude faster evaluations on a large suite of diverse benchmarks.
2. The large-scale analysis of 13 ZC proxies across 28 different combinations of search spaces and tasks reveals the generalizability, bias, and mutual information among ZC proxies.
3.  As a kind of complementary information,  ZC proxies can significantly improve the predictive power of surrogate models commonly used for NAS
4. The paper is easy to read, and in terms of the open source code, it provides an easy-use interface and well-organized documentation.

**Weaknesses:**

1. It is largely based on NAS-Bench-Suite, and the main difference comes from more datasets, publicly releasing ZC proxy values, combining ZC proxies in a nontrivial way, and exploiting the complementary information of 13 ZC proxies simultaneously.
2. The in-depth analysis from Section4 could have been more as the basic information largely overlapped with NAS-Bench-Suite. The bias part in Section 4.3 is interesting, perhaps more analysis in the relation between the design principle of zero-cost proxy and results.
3. The definition and explanation in the table content can be more detailed such as in Table3.



**Additional Feedback:**

Please check the Weakness part.

**Correctness:**

The claims made in the submission look correct. The evaluation methods and experiment design are appropriate and performed correctly.

**Documentation:**

The required information is well organized and easy to access.

**Ethics:**

I did not see any ethical concerns.

**Relation To Prior Work:**

It mainly extends the zero-cost part of NAS-Bench-Suite. The difference from the previous contributions is clearly discussed.

**Summary And Contributions:**

In this work, NAS-Bench-Suite-Zero evaluates 13 zero-cost proxies for architecture performance prediction across 28 tasks. It creates by far the largest dataset, enabling orders-of-magnitude faster experiments on ZC proxies. The provided codebase is accessible through a unified interface, created with the aim to facilitate reproducible, generalizable, and rapid NAS research.

---

> ### Author Response · Authors · 2022-08-14
> **Thank you for the insightful feedback**
>
> We thank the reviewer for their thorough and insightful review.
>
> **"Based on NAS-Bench-Suite"**
>
> The groundwork is indeed based on NAS-Bench-Suite, but as the reviewer themselves mentions, the difference comes from publicly releasing 1.5M ZC proxy values, combining ZC proxies in a nontrivial way, exploiting their complementary information, and running additional analyses.
>
> **"Analysis from Section 4.3 could have been more"**
>
> This is a great suggestion. We have now included a concrete method for reducing bias in ZC proxies. Please see Appendix D.3.1. We find that we can (1) reduce bias to zero at the cost of performance, or (2) reduce bias so that it matches the bias of the ground truth validation accuracy, which often improves performance.
> Although there are still a few details before this can be efficient (e.g., how to quickly determine the bias of the ground truth architectures), we believe this can have important ramifications for future ZC proxy design. Thank you for this comment!
>
> **"Table content can be more detailed"**
>
> We agree, and we have now increased the captions of Tables 3, 5, 6, and others.
>
> We thank the reviewer once again for these excellent suggestions.

---

### Official Review · Reviewer_mnRg · 2022-07-28
**Comprehensive evaluation on an under-studied area**

**Rating:** 7
**Confidence:** 4
**Clarity:** The paper is well structured and writ…

**Strengths:**

- A good summary and comprehensive comparison of zero-cost proxies published.
- Emprical analysis of the performance of many zero-cost proxies. A deep dive into the generalization abilities of them on different search spaces and tasks. This kind of analysis is not done before.
- Understanding if the zero-cost proxies can explain the ground truth accuracy, and how they complement each other (in terms of information gain).
- A novel way to combine all zero-cost proxies studied as features into the NAS predictor.

**Weaknesses:**

- Most results are empirical which provide some valuable insight into so many ZC proxies. However, there is no theorethical analysis or explaination on such observations.


**Additional Feedback:**

- The greedy approach is not visible in Figure 4?

**Correctness:**

To the best of my knowledge the claims made are correct. The dataset is constructed property with detailed documentation and code repository. The evaluation methods are well designed.

**Documentation:**

The dataset is well documented and hosted on a public repository for access.

**Ethics:**

No ethical concern.

**Relation To Prior Work:**

Section 2 mentioned predictor-based NAS methods but only [46] is cited. There are more related work:
* BRP-NAS: Prediction-based NAS using GCNs. https://arxiv.org/abs/2007.08668
* Bridging the gap between sample-based and one-shot neural architecture search with bonas. https://arxiv.org/abs/1911.09336
* Bananas: Bayesian optimization with neural architectures for neural architecture search. https://arxiv.org/abs/1910.11858

**Summary And Contributions:**

This paper presents the analysis of 13 zero-cost proxies on 28 tasks, and releases the pre-computed scores of them. The authors also studied a way to combine all zero cost proxies to allow better NAS performance of predictior-based algorithms.

---

> ### Author Response · Authors · 2022-08-14
> **Thank you for your review**
>
> We thank the reviewer for their insightful review. We respond to the points below.
>
> **"No theoretical analysis"**
>
> We agree that theoretical analysis is very beneficial to this field. However, since this is the only point the reviewer lists as a weakness, we respectfully remind the reviewer that this is a track that focuses on datasets and benchmarks, and as such theory may be less of an emphasis. Nevertheless, we have now added a section to our paper (Appendix C.1) that surveys the theoretical analysis for ZC proxies, which we think will be very helpful for readers. We thank the reviewer for bringing this to our attention.
>
> **"No explanations"**
>
> We respectfully point out that many of our empirical findings do have explanations, and we will make this more explicit in the paper. For example, the explanation for the improved NAS experiments (Section 5) are partly explained by complementary information (Section 4.2), and the relative performance trends of ZC proxies (Section 4.1) are partly due to biases (Section 4.3). We also just added feature importance scores for the surrogate models (Appendix E.1), which helps to explain our results even more.
>
> **"Additional related work"**
>
> We thank the reviewer for pointing out these related works. We have now updated the paper to include them in Section 2.
>
> **"Figure 4"**
>
> The minimum k tuple and greedy approaches tie in the first two images (we briefly mentioned this in the caption but we can make this clearer).
>
> If the reviewer feels their concerns are addressed, we respectfully ask the reviewer to consider increasing their score. If you have any more questions, please let us know and we would be happy to continue the discussion.

---

> > ### Comment · Reviewer_mnRg · 2022-08-24
> > **Thank you for the response**
> >
> > Thank you. I am satisfied with the additional details and clarifications. I have updated the score.

---

### Author Response · Authors · 2022-08-14
**Revised paper following reviewers’ comments**

Dear reviewers and AC, we have now addressed all of the suggestions and concerns mentioned by the reviewers. We thank the reviewers very much for these comments, which we believe has substantially improved our work.

The full list of changes in the new version of our paper are as follows
- Our documentation in Appendix B was already described as a strength by the reviewers. Following reviewer feedback, for better visibility, we now include all documentation directly in the [readme](https://github.com/automl/NASLib/tree/zerocost#documentation), including our
[datasheet](https://github.com/automl/NASLib/blob/zerocost/docs/DATASHEET.md),
[license](https://github.com/automl/NASLib/blob/zerocost/LICENSE),
[author responsibility](https://github.com/automl/NASLib/blob/zerocost/docs/AUTHOR_RESPONSIBILITY.md),
[code of conduct](https://github.com/automl/NASLib/blob/zerocost/docs/CODE_OF_CONDUCT.md), and
[maintenance plan](https://github.com/automl/NASLib/blob/zerocost/docs/MAINTENANCE_PLAN.md).
- We added **initial results for FBNet** (which includes MobileNet) as a new search space in NAS-Bench-Suite-Zero. Please see Appendix D.1.1.
- We present a **concrete method to reduce the bias of zero-cost proxies** (Appendix D.3.1), which may have important consequences for the future design of ZC proxies.
- We present an analysis of feature importance of the ZC proxy features on the trained surrogate mode (Appendix E.1).
- We give an ablation on the discretization strategy for the information theoretic analysis (Appendix D.2).
- We re-run Figure 2 using additional metrics: Precision@K, BestRanking@K (Appendix D.1).
- We give new information gain experiments that target the top-k architectures (Appendix D.2).
- We release a new quickstart guide for using the NAS-Bench-Suite-Zero dataset (see [here](https://github.com/automl/NASLib/blob/zerocost/plotting/PlotCorrelations.ipynb)).
- We add a new section that surveys theoretical results for ZC proxies (Appendix C.1).

We would also like to mention that our benchmark suite was very recently used successfully in a competition – the [Zero Cost NAS Competition at AutoML-Conf 2022](https://sites.google.com/view/zero-cost-nas-competition/home). During the competition, participants used our benchmark suite to develop new, better versions of ZC proxies. This is additional evidence that our work can be used by the community to develop better NAS techniques. We will include more details of the competition in our paper.

We thank all reviewers once again for the above suggestions. We are happy to address any new follow-ups or concerns.

---

> ### Author Response · Authors · 2022-08-22
> **Additional update**
>
> Dear reviewers and AC, we have now updated our paper with two more additions
> - Ablation study on the number of ZC proxies vs Spearman rank correlation, for the surrogate model (see Appendix E.2). The best performance is achieved with all 13 ZC proxies.
> - We included a section (Appendix F) on the ZC proxy competition that was held using NAS-Bench-Suite-Zero. During the competition, participants used our repository to develop better versions of synflow, fisher, and grad_norm.
>
> We would also like to again highlight three changes from our [update from Aug 14](https://openreview.net/forum?id=yWhuIjIjH8k&noteId=W0wYK_SGWxh):
> - We added all of our dataset documentation to the [readme](https://github.com/automl/NASLib/tree/zerocost#documentation), so it is now in the readme and appendix.
> - We added a new search space to NAS-Bench-Suite-Zero: FBNet (which includes MobileNet).
> - We added a new concrete method to mitigate the biases of ZC proxies, and we demonstrated that it can be used to improve the accuracy of ZC proxies.
>
> Thank you very much Reviewer A4Zd for responding to our rebuttal. If any of the other reviewers have additional questions, comments, or clarifications, we would be very happy to give a reply before the response period ends on August 29. Thank you!

---

### Meta-Review · Area_Chair_Zaak · 2022-09-12

**Recommendation:** Accept
**Confidence:** 4

**Metareview:**

This paper presents a systematic review of zero-cost proxies for neural architecture search. By design, these proxies are cheap and easy-to-evaluate but results reported in the literature have been mixed. In this light, a benchmark seems relevant and important for further progress in this area. The reviewers uniformly appreciated the extent of tasks and proxies considered. Moreover, the reviewers also appreciated efforts made to explore the complementary strengths of these proxies. There were some concerns related to the search space and metrics used, but those were addressed by the authors during the rebuttal period. Finally, the availability of open-sourced and well-documented code is a plus that was appreciated by everybody.

---

### Decision · Program_Chairs · 2022-09-16

Accept